# Evolution of binding preferences among whole-genome duplicated transcription factors

**Tamar Gera†, Felix Jonas†, Roye More, Naama Barkai\***

Department of Molecular Genetics, Weizmann Institute of Science, Rehovot, Israel

**Abstract** Throughout evolution, new transcription factors (TFs) emerge by gene duplication, promoting growth and rewiring of transcriptional networks. How TF duplicates diverge was studied in a few cases only. To provide a genome-scale view, we considered the set of budding yeast TFs classified as whole-genome duplication (WGD)-retained paralogs (~35% of all specific TFs). Using high-resolution profiling, we find that ~60% of paralogs evolved differential binding preferences. We show that this divergence results primarily from variations outside the DNA-binding domains (DBDs), while DBD preferences remain largely conserved. Analysis of non-WGD orthologs revealed uneven splitting of ancestral preferences between duplicates, and the preferential acquiring of new targets by the least conserved paralog (biased neo/sub-functionalization). Interactions between paralogs were rare, and, when present, occurred through weak competition for DNA-binding or dependency between dimer-forming paralogs. We discuss the implications of our findings for the evolutionary design of transcriptional networks.

## Editor's evaluation

The authors use creative and innovative approaches to explore the evolution of transcription factors following duplication. This paper will be broad interest among evolutionary and molecular biologists as it addresses the long-standing question of how newly evolved transcription factor proteins acquire new binding specificities or split ancestral ones. Among the major findings, the authors find the changes in binding specificity occur mainly through changes outside of the DNA binding domains, showing that novelties in transcription factor binding can occur through multiple routes following gene duplications.

**\*For correspondence:**
naama.barkai@weizmann.ac.il

†These authors contributed equally to this work

## Introduction

Transcription factors (TFs) bind at regulatory regions to activate or repress transcription. Cells express hundreds of TFs that together encode a variety of expression programs. Despite rapid advances, our understanding of transcriptional network design is still fragmented (*Jana et al., 2021*). For example, different TFs that bind to similar DNA sequences *in vitro* (*Weirauch et al., 2014*; *Wei et al., 2010*; *Jolma et al., 2013*; *Matys et al., 2006*; *Siggers et al., 2014*; *Nakagawa et al., 2013*; *Rogers and Bulyk, 2018*; *Berger et al., 2008*; *Shen et al., 2018*) localize to different genomic sites *in vivo* through poorly understood mechanisms. Further, with increasing organism complexity, new TFs emerge, yet we know little about how these emerging TFs adapt new targets and integrate into the existing transcriptional network.

Gene duplication is the major source of new TFs (*Ohno et al., 1968*; *Rosanova et al., 2017*; *Vaquerizas et al., 2009*; *Levine and Tjian, 2003*; *Teichmann and Babu, 2004*), with whole-genome duplications (WGDs) playing a particularly important role (*Blanc and Wolfe, 2004*; *Maere et al.,*

*2005*; *Rody et al., 2017*; *Wolfe and Shields, 1997*; *Marcet-Houben and Gabaldón, 2015*; *Birchler et al., 2005*; *Freeling and Thomas, 2006*; *Edger and Pires, 2009*; *Lundin, 1993*; *Dehal and Boore, 2005*; *Blomme et al., 2006*). In budding yeast, ~35% of all TFs are associated with a single WGD event dating back to ~100 million years ago (*Figure 1A*; *Wolfe and Shields, 1997*; *Marcet-Houben and Gabaldón, 2015*). We reasoned that this set of TF duplicates, all generated at the same time and subjected to the same evolutionary history, provides a convenient platform for studying the fate of duplicated and retained TF genes.

TF duplicates (paralogs) can diverge through changes in expression, regulation, or function (*Vavouri et al., 2008*; *DeLuna et al., 2008*; *Diss et al., 2014*; *Kafri et al., 2006*; *Wapinski et al., 2007*; *Gu et al., 2003*; *VanderSluis et al., 2010*; *Charoensawan et al., 2010*; *Des Marais and Rausher, 2008*; *Hsiao and Vitkup, 2008*; *Ihmels et al., 2007*; *Kuzmin et al., 2020*; *Ehrenreich, 2020*; *Payne and Wagner, 2015*; *Chen et al., 2013*; *Burga et al., 2011*; *Macneil and Walhout, 2011*; *Diss et al., 2017*). Of particular interest is the TF's selection of *in vivo* binding sites, as these define potential regulatory targets. Mechanisms driving divergence of TF-binding sites include changes in co-factor binding (*Baker et al., 2013*) or in DNA motif preferences (*Pérez et al., 2014*; *Blake and Barolo, 2014*; *McKeown et al., 2014*; *Bridgham et al., 2008*; *Humbert et al., 2013*; *Pougach et al., 2014*). The prevalence of these different scenarios is still unclear, since the studied cases considered only a few paralogs, of different ages and origins, and TF binding was measured at individual targets. We therefore aimed to provide a genome-scale view, by comparing genome-wide binding preferences among the full set of WGD TFs in budding yeast.

## Results

### Divergence of binding preferences among WGD TFs

Within the *Saccharomyces cerevisiae* genome database (SGD) (*Cherry et al., 2012*), 82 proteins containing an annotated DNA-binding domain (DBD) are classified as WGD-retained paralogs (*Figure 1A and B*). We refined this list to include only pairs where both paralogs act as specific TFs (35 pairs, *Supplementary file 1*), and defined the binding locations of these TFs across the genome using chromatin endogenous cleavage with high-throughput sequencing (ChEC-seq) (*Zentner et al., 2015*). A total of 30 pairs (60 TFs) were successfully profiled, as verified by data reproducibility (Pearson's r>0.95 in promoter-binding preferences, *Figure 1C*) and manual literature survey (*Figure 1—figure supplement 1*). A large fraction of TFs were bound at their own or their paralog's promoters, potentially forming regulatory circuits (*Figure 1E*; *Teichmann and Babu, 2004*).

Perhaps unexpectedly, binding preferences were conserved (Pearson's r>0.8) among ~40% of paralogs, most notably these of the C2H2 zinc finger family (e.g. Met31/Met32, *Figure 1D*, 6/10 pairs *Figure 2*). Furthermore, most diverging paralogs still shared a substantial fraction of target promoters (*Figure 2*). In some cases, the two duplicates localized to the same promoters but with different relative strengths (e.g. Gzf3/Dal80, *Figure 1D*). In other cases, some promoters were bound by both paralogs, while others were preferentially bound by just one paralog (e.g. Ace2/Swi5, *Figure 1D*). Therefore, binding preferences diverge at a rate that differs between pairs, and, within each pair, differs between individual promoters.

### Paralogs diverge through variations outside their DBDs

TFs localize to genomic sites containing short motif sequences bound by their DBDs. The *in vivo* binding could therefore diverge through DBD variations that modify motif preferences. To compare DBD sequences among paralogs, we aligned each pair and classified residues into these defining the DBD family (e.g. C/H residues in the C2H2 zinc finger domains), those that contribute to DNA motif preferences (*Lambert et al., 2019*), and the remaining ones (*Figure 3A, B* and *Figure 3—figure supplement 1*).

Sequence conservation varied between DBD families (*Figure 3B*). In particular, specificity-conferring residues often varied between paralogs of the fungal-specific zinc cluster family (*Figure 3B*), but remained invariant between paralogs of the C2H2 zinc finger family (e.g. Rph1/Gis1) and, to a lesser extent, other families (e.g. Dot6/Tod6, *Figure 3A and B*). Examining motif preferences derived from *in vitro* data, we noted that reported preferences (*Weirauch et al., 2014*; *Lambert et al., 2019*) are often (although not always) similar among paralogs (*Figure 3A* and *Figure 3—figure supplement 1*).

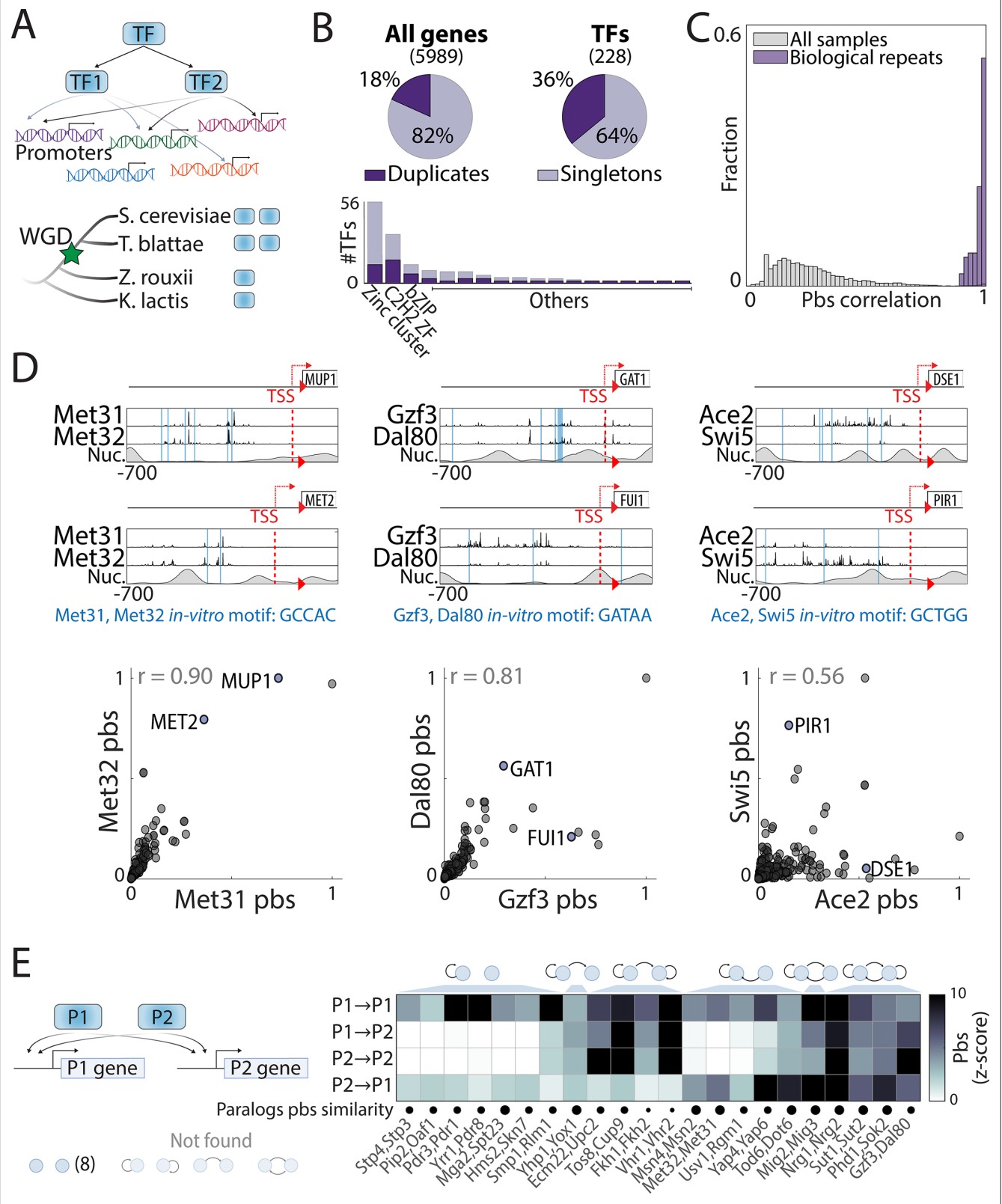

**Figure 1.** Mapping the promoter-binding preferences of whole-genome duplication (WGD) transcription factors (TFs). (**A–B**) *WGD shaped the budding yeast transcription network:* (**A**) TF duplicates (paralogs) can diverge to bind different targets. (**B**) In *Saccharomyces cerevisiae*, ~35% (*Gietz et al., 1995*) of all present-day TFs are retained WGD paralogs, belonging to 18 different DNA-binding domain families (see *Figure 1—figure supplement 1*). (**C**) *TF-binding profiles are reproducible*: Shown is the distribution of correlations between different samples (gray) and between biological repeats

*Figure 1 continued on next page*

*Figure 1 continued*

(purple). Correlations are between promoter-binding signals (pbs). (**D**) *Binding profiles of indicated TF-paralog pairs.* Top: Measured binding signal and nucleosome occupancy (Nuc.) on individual promoters (see Materials and methods). Lines indicate transcription start sites (TSS, red dashed) and locations of *in vitro* motifs (blue). Bottom: Pbs of the indicated TF-paralog pairs (each dot is a promoter, r: Pearson's correlation). (**E**) *Auto- and cross-promoter binding by TF paralogs*: Pbs is shown as z-score. Potentially formed circuits indicated on top. Note that 22/30 pairs are associated with six of nine possible circuits.

The online version of this article includes the following source data and figure supplement(s) for figure 1:

**Source data 1.** Details for statistical analysis (correlations).

**Figure supplement 1.** Sensitive, accurate, and reproducible mapping of whole-genome duplication (WGD) transcription factors (TFs) DNA-binding profiles with chromatin endogenous cleavage with high-throughput sequencing (ChEC-seq).

DBD variations may therefore contribute to the divergence of zinc cluster paralogs, but appear to play a lesser role in paralogs of the C2H2 zinc finger and perhaps other families. To test this, we swapped DBDs between paralogs (*Figure 4A*), reasoning that if DBD variations drive divergence, swapping DBDs would swap promoter preferences. Conversely, if critical variations are located outside the DBD, swapping will be of little effect. Consistent with their strong DBD sequence divergence, DBD-swapping perturbed promoter binding for three of the four zinc cluster TFs tested. However, in none of these was DBD-swapping sufficient for switching promoter preferences to those of the paralog from which the DBD was taken (*Figure 4B and C*). Further, in all other 12 cases studied, binding preferences remained largely invariant to the swapping of the DBD (Pearson's r>0.8). Of note, this invariance to DBD-swapping was also observed when comparing *in vivo* 7-mer DNA sequence preferences (*Figure 4B* and *Figure 4—figure supplement 1*). We conclude that, for most paralog pairs, the variations driving divergence in promoter-binding preferences are located outside the DBDs.

## Dependencies and competitions between TF paralogs

Evolved interactions between paralogs could affect binding preferences. Paralog TFs that bind DNA as dimers, for example, may bind as heterodimers to a subset of sites. Paralogs may also compete for DNA binding, either directly or by interacting with a shared co-factor. In the broader context, cooperative interactions, where a TF becomes dependent on its paralog, may increase mutation fragility whereas binding competition, which allows a TF to access sites bound exclusively by its paralog upon the latter's perturbation, may increase mutation robustness (*Figure 5A*). Both effects were observed in the context of protein-protein interactions (*Diss et al., 2017*). To test the prevalence of cooperative or competitive interactions, we measured TF binding upon paralog deletion, testing 55/60 TFs in our dataset (*Figure 5B* and *Figure 5—figure supplement 1*). Two TFs completely lost binding signals (Pip2, Hms2) upon paralog deletion, and additional two lost binding to their respective paralogs' targets (Dal80, Tbs1). These large-scale effects, however, were infrequent (*Figure 5B and C*). In fact, cooperative interactions were generally minor (e.g. Stp2), as were compensatory interactions (e.g. Pdr3 or Ecm22; *Figure 5C and D*). Therefore, strong interactions between TF paralogs are rare and existing ones tend to increase mutation fragility.

## Classifying paralogs' evolutionary paths by analyzing non-WGD orthologs

Two prevailing models are commonly invoked to explain paralog divergence (*Voordeckers et al., 2012*; *Lynch and Conery, 2000*; *Sugino and Innan, 2006*; *Lynch and Force, 2000*; *Conant et al., 2006*; *Force et al., 1999*; *Kondrashov et al., 2002*; *He and Zhang, 2005*; *Nowick and Stubbs, 2010*): In the first, ancestral functions split between duplicates (sub-functionalization) while, in the second, one duplicate retains ancestral functions, while the other adapts a new role (neo-functionalization, *Figure 6A*). As a first approach to distinguish these two scenarios, we used phylogenetic sequence analysis to compare the evolutionary rates of regions outside the conserved DBDs (*Figure 6B–D*). This analysis is informative, since a neo-functionalizing paralog undergoes a period of relaxed selection, followed by rapid evolution, and is therefore expected to evolve at an accelerated rate (*Pegueroles et al., 2013*; *Byrne and Wolfe, 2007*). We observe that paralogs of the C2H2 zinc finger family, including the diverging ones, evolved symmetrically, that is, at rates that did not distinguish between paralogs, consistent with sub-functionalization (e.g. Tda9/Rsf2 and Gis1/Rph1, *Figure 6C and D*).

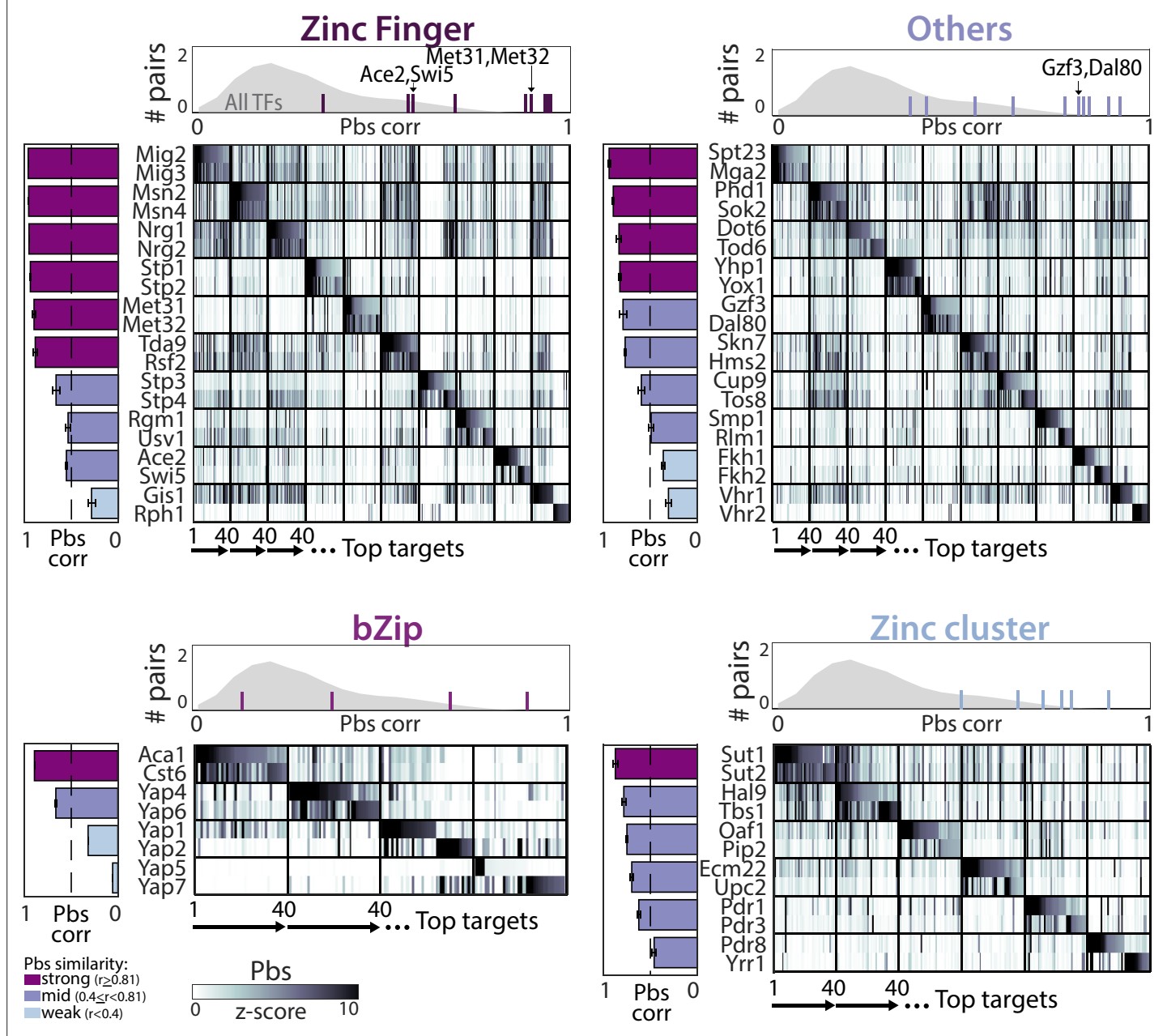

**Figure 2.** Divergence of promoter-binding preferences in whole-genome duplication (WGD) transcription factor (TF) paralogs. The 40 top-bound promoters by each paralog pair (y-axis) were selected (see *Supplementary file 2*), ordered along the x-axis, and color-coded according to promoter binding signal (pbs, z-score). TFs are organized by DNA-binding domain (DBD) families, as indicated. Bars on the left depict correlations in binding preferences (pbs similarity) of respective paralogs and are summarized for all paralogs of the indicated family (individual lines) and non-paralog TFs (gray) in the histogram on top.

By contrast, diverging paralogs of other families evolved asymmetrically, suggesting dominant neo-functionalization (e.g. Vhr1/Vhr2, *Figure 6C and D*).

To test experimentally for sub- and neo-functionalization, we compared binding preferences of 18 paralog pairs to that of a corresponding non-WGD ortholog (*Kluyveromyces lactis* TF expressed within *S. cerevisiae*; *Figure 7A*). We reasoned that, in terms of binding preferences, this non-WGD ortholog might serve as a good proxy for the ancestor TF (*Teichmann and Babu, 2004*; *Hsia and McGinnis, 2003*; *Carroll, 2005*). This was the case in pairs with clear expectations; the Ixr1 and Abf2 paralogs have diverged completely, with Abf2 having a mitochondrial function and localization (*MacPherson*

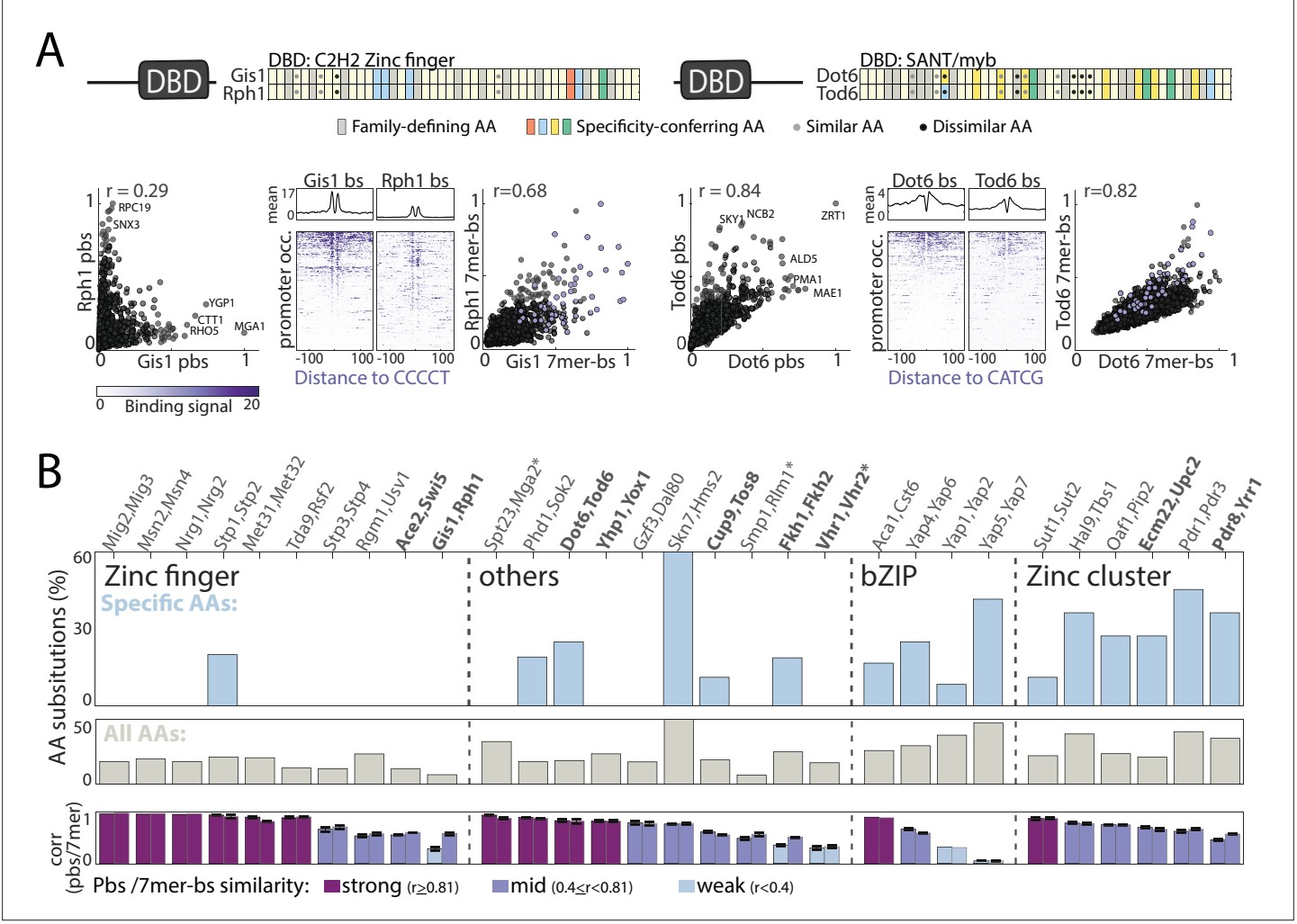

**Figure 3.** Sequence variations distinguishing paralogs' DNA-binding domains (DBDs). (**A–B**) *Classifying DBD residue substitutions*. (**A**) For all paralog pairs, Pfam-defined DBDs were aligned and residues classified into those conserved among all family members (gray) and specificity-conferring ones (colored: blue, red, yellow, and green denoting positive, negative, hydrophobic and hydrophilic residues, respectively) (*Lambert et al., 2019*). Amino acid (AA) substitutions into biophysically similar or dissimilar residues are indicated as gray and black dots, respectively. Two examples are shown, as indicated, see *Figure 3—figure supplement 1* for other pairs. Also shown are comparisons of binding signals across promoters (left) or 7-mers (right, purple dots indicate 7-mers containing the *in vitro* motif), as well as the binding signals around *in vitro*-motif occurrences (middle). (**B**) Fraction of amino acid substitutions among specificity-conferring (top) and all (middle) DBD residues between the paralog pairs. Also shown are the correlations in promoter and 7-mer binding signal between the respective paralogs (bottom, left and right bar, respectively). Note the little correspondence between DBD sequence variations and divergence of binding profiles. Paralogs chosen for further DBD-swapping analysis (*Figure 4*) are highlighted in bold, *: indicates paralogs from DBD families where specificity-conferring residues are not available.

The online version of this article includes the following figure supplement(s) for figure 3:

**Figure supplement 1.** Sequence variations distinguishing paralog DNA-binding domains (DBDs).

et al., 2006) and Ixr1 being a nuclear repressor regulating hypoxia genes. We find that the binding preferences of Ixr1/Abf2's *K. lactis* ortholog were indistinguishable from these of Ixr1 (*Figure 7A*), consistent with Abf2's accelerated sequence evolution (*Figure 6B*). Similarly, *K. lactis* orthologs of paralogs with non-diverged binding preferences retained highly similar preferences, for example, Rsf2/Tda9 regulators of respiration-related functions (*Figure 7B and C* and *Figure 7—figure supplement 1* for additional pairs). Together, these results support the use of non-WGD orthologs to approximate ancestral preferences.

We next extended the analysis to divergent duplicates. We observed cases of clear sub- and neo-functionalization (Rph1/Gis1 and Vhr1/Vhr2, respectively, *Figure 7B*), but most pairs showed a combination of the two scenarios. Of note, in all 11 diverging cases, binding preference similarity of at least

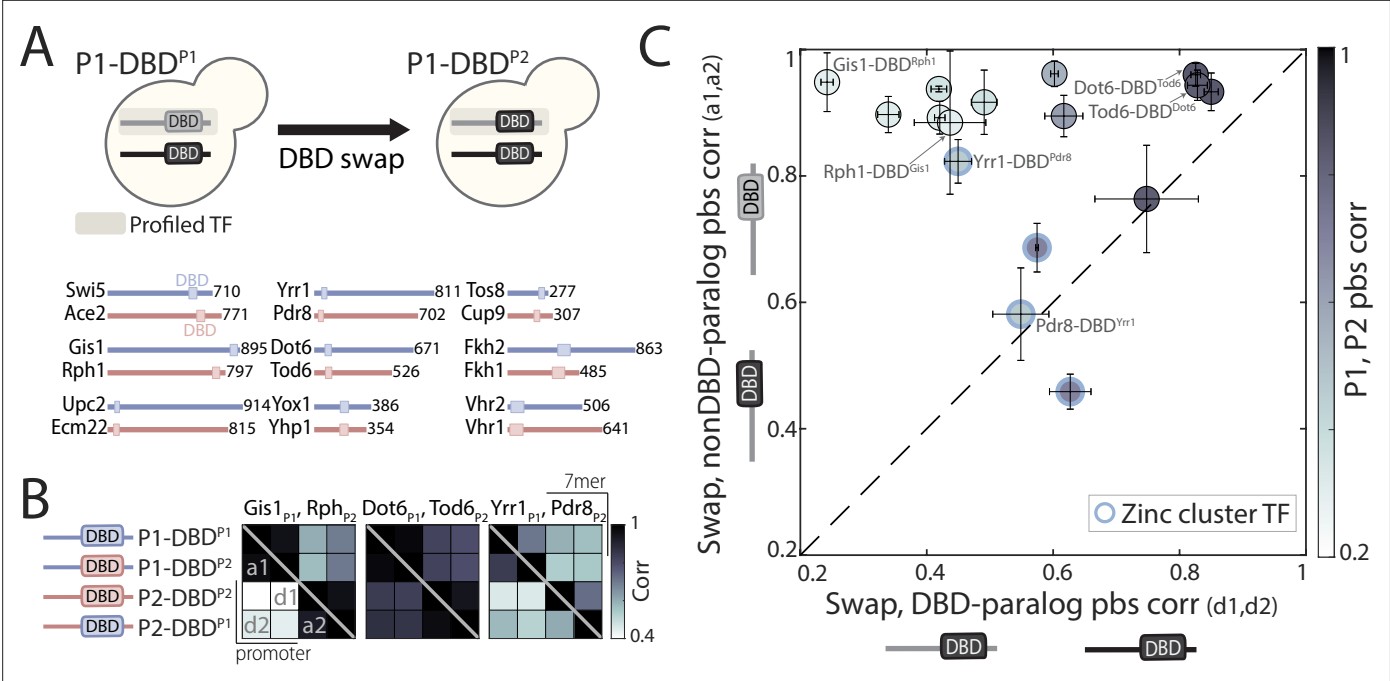

**Figure 4.** DNA-binding domain (DBD) swapping has a minor effect on binding preferences. (**A**) *DBD-swapping experimental scheme:* DBDs of the indicated paralog pairs were swapped, and their binding profiles mapped. (**B**) Correlations of binding preferences between the indicated transcription factors (TFs) and their swapped variants (bottom triangle: promoters, top triangle: 7-mers; see also *Figure 4—figure supplement 1* for all tested pairs). (**C**) Correlation in promoter-binding signals (pbs) between paralogs and their swapped variants, as indicated. Blue indicates zinc cluster TFs; shading depicts correlation between the wild-type paralogs. Note that outside the zinc cluster family, DBD-swapping is of little consequence for promoter-binding preferences, even among highly divergent paralogs. Within the zinc cluster family, DBD-swapping affected binding profiles, but did not recover binding preferences of the paralog from which the DBD was taken.

The online version of this article includes the following source data and figure supplement(s) for figure 4:

**Source data 1.** Details for statistical analysis (correlations).

**Figure supplement 1.** Swapping experiment confirms functional conservation of DNA-binding domains (DBDs) between paralogs.

one paralog with the *K. lactis* ortholog exceeded the similarity with the other *S. cerevisiae* paralog. In nine of these cases, the sequence of the paralog with the more conserved binding profile was also slower to evolve (*Figure 7C*). The prevalent route for diversification therefore appears to comply with a biased neo/sub-functionalization: ancestral preferences split unevenly between the duplicates, coupled to biased acquisition of novel targets (*Figure 7D* and *Figure 7—figure supplement 1*).

## Divergence of zinc cluster paralogs: motif preferences and paralog interactions

To gain molecular insights into the divergence of specific TF duplicates, we examined individual cases, focusing first on the fungal-specific zinc cluster family. TFs of this family bind DNA as dimers, recognizing a composite DNA motif that includes two spaced CGG sites (*MacPherson et al., 2006*). Binding specificity depends on the orientation of the DBD-bound CGG triplets, and on the spacer length, which likely relates to the unstructured linker flanking the DBD (*Figure 8A*; *Reece and Ptashne, 1993*). Compared to other families, divergence of zinc cluster paralogs was more dependent on DBD variations (*Figure 4C*), and these paralogs were more likely to interact (*Figure 5*).

We searched for CGG-containing motifs in proximity to the zinc cluster TF-binding sites in our data. In three of the five diverging pairs, differences in binding preferences correlated with differential motif preferences (*Figure 8A*, *Figure 8—figure supplement 1*): Hal9/Tbs1 and Oaf1/Pip2 showing differential spacing/orientation and Pdr1/Pdr3 showing variation within the CGG. All three pairs interacted: Hal9/Tbs1 recruited each other to their preferred sites, likely through heterodimerization, and Pip2 localized exclusively to a subset of Oaf1-preferred targets in an Oaf1-dependent manner, consistent with obligatory heterodimerization (*Rottensteiner et al., 1997*). Notably, binding preferences of both

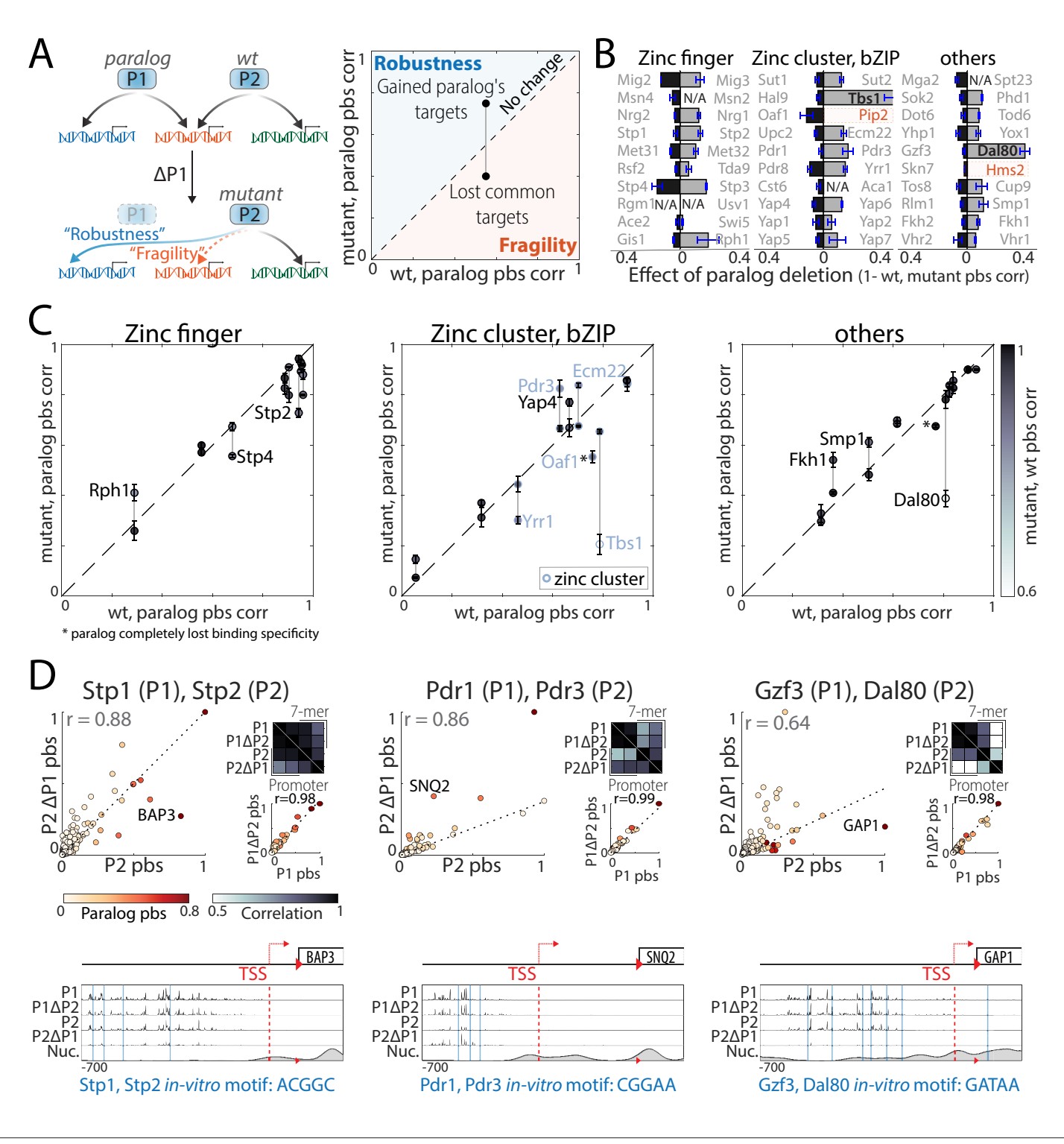

**Figure 5.** Interactions between transcription factor (TF) paralogs may increase network fragility. (**A**) *Paralogs' contribution to mutation robustness or fragility:* Following paralog deletion, a TF may gain access to its paralog's unique sites, potentially compensating for the loss ('robustness', blue line: gained paralog target). Alternatively, paralogs may become interdependent and loose common targets after paralog deletion ('fragility', dashed orange line: lost common target). At the genome level, these interactions can be summarized by comparing a TF's binding preferences in wild-type (x-axis) or paralog-deleted (y-axis) backgrounds to those of the paralog. (**B**) *Strong paralog interactions are rare*: the effect of paralog deletion on promoter-binding preferences was measured for 55 of 60 TFs in our dataset. Shown is the effect of paralog deletion on binding preferences for each TF. Note

*Figure 5 continued on next page*

*Figure 5 continued*

that most deletions were of little effect and that large effects were asymmetric. Also indicated are substantial effects (TFs written in black) and TFs that completely lost binding specificity (orange, see Materials and methods; N/A: not profiled). (**C–D**) *Paralog interactions within individual families*: (**C**) robustness/fragility analysis, as in (**A**) for all tested paralog pairs, divided into families (*: paralog completely lost binding specificity). (**D**) Shown are individual examples of the depicted correlations (see *Figure 5—figure supplement 1* for all tested pairs). Note that Stp2 and Dal80 loose binding to some of their paralog's targets upon paralog deletion ('fragility'), whereas Pdr3 gains binding to Pdr1 targets (e.g. SNQ2) upon the latter's deletion ('robustness').

The online version of this article includes the following source data and figure supplement(s) for figure 5:

**Source data 1.** Details for statistical analysis (correlations).

**Figure supplement 1.** Paralog deletion indicates gene-specific paralog-paralog interactions.

heterodimers correlated well with that of the *K. lactis* ortholog (*Figure 8A*). In the case of Pdr1/Pdr3, interaction took the form of competition, with Pdr1 outcompeting Pdr3 from accessing its preferred binding sites (*Figure 8A*).

In these three cases described above, binding preferences evolved through a combination of DNA-motif preferences and protein-protein interactions. Contrasting these, CGG-containing motifs at Yrr1/Pdr8- or Upc2/Ecm22-bound sites did not distinguish between paralogs (*Figure 8A–D* and *Figure 8—figure supplement 1*). Rather, paralogs localized to the same motifs, but within different promoters. This divergence of promoter preferences was largely DBD-independent in the case of Yrr1/Pdr8 (*Figure 4*). In the case of Upc2/Ecm22, on the other hand, it was largely DBD-dependent competition (*Figure 8D*). In fact, upon UPC2 deletion or DBD-swapping, Ecm22 gained access to strong Upc2 sites (*Figure 8C–E*; note correspondence to *K. lactis* sites). Of note, while DBD-swapping retained binding at the TF-specific sites, it also allowed increased access to non-specific sites, suggesting co-evolution of the DBD and the linker domain, both of which varied substantially between the paralogs (*Yang et al., 2015*). We conclude that zinc cluster paralogs evolved largely, but not exclusively, through changes in motif preferences or affinity, resulting from combined effects of variations within, and outside, their DBDs.

## Resolution of paralog interference through competitive binding

In the case of the zinc cluster paralogs, divergence of motif preferences has played a major role in the evolutionary scenarios governing their divergence. In most other paralogs, changes in motif preference, if exist, appear to be secondary to the major effective variations located outside the DBD. Still, even in such cases, DBD variations may play a role in resolving residual paralog interference (*Baker et al., 2013*; *Kaltenegger and Ober, 2015*). Thus, since the two paralogs are initially redundant, divergence of binding preferences requires not only the acquisition of differential preferences, but also limiting residual, and possibly interfering paralog's cross binding. In the case of the non-WGD pair Mcm1/Arg80, for example, such interference was resolved by weakening direct Arg80-DNA binding (*Baker et al., 2013*; *Kaltenegger and Ober, 2015*). We asked whether similar effects are observed for WGD paralogs.

DBD variations contributed little to the divergence of Rph1/Gis1, the most diverged C2H2 zinc finger paralogs. We noted, however, that Rph1 gained residual access to Gis1-binding sites upon GIS1 deletion or DBD-swapping (*Figure 9A and B*). Further, Rph1-exclusive targets contained a specific variant of the common Gis1/Rph1 motif, lacking from Gis1-binding sites (*Figure 9C and D*). Phylogenetic analysis revealed two sequence variations between the paralogs: first, Gis1 has lost a conserved demethylase activity (*Klose et al., 2006*), an event that occurred quite soon following WGD. Second, Rph1's DBD experienced a more recent (minor) variation altering a conserved residue (*Figure 9E* and *Figure 9—figure supplement 1*). Together, this suggests the following evolutionary scenario: first, a loss of demethylase activity allowed Gis1 to specialize toward a subset of (weaker) ancestral targets and acquire new binding sites, through a primarily DBD-independent evolution. Second, mutations within Rph1's DBD prevented its binding to Gis1-specialized sites, thereby reducing paralog interference (*Figure 9E*).

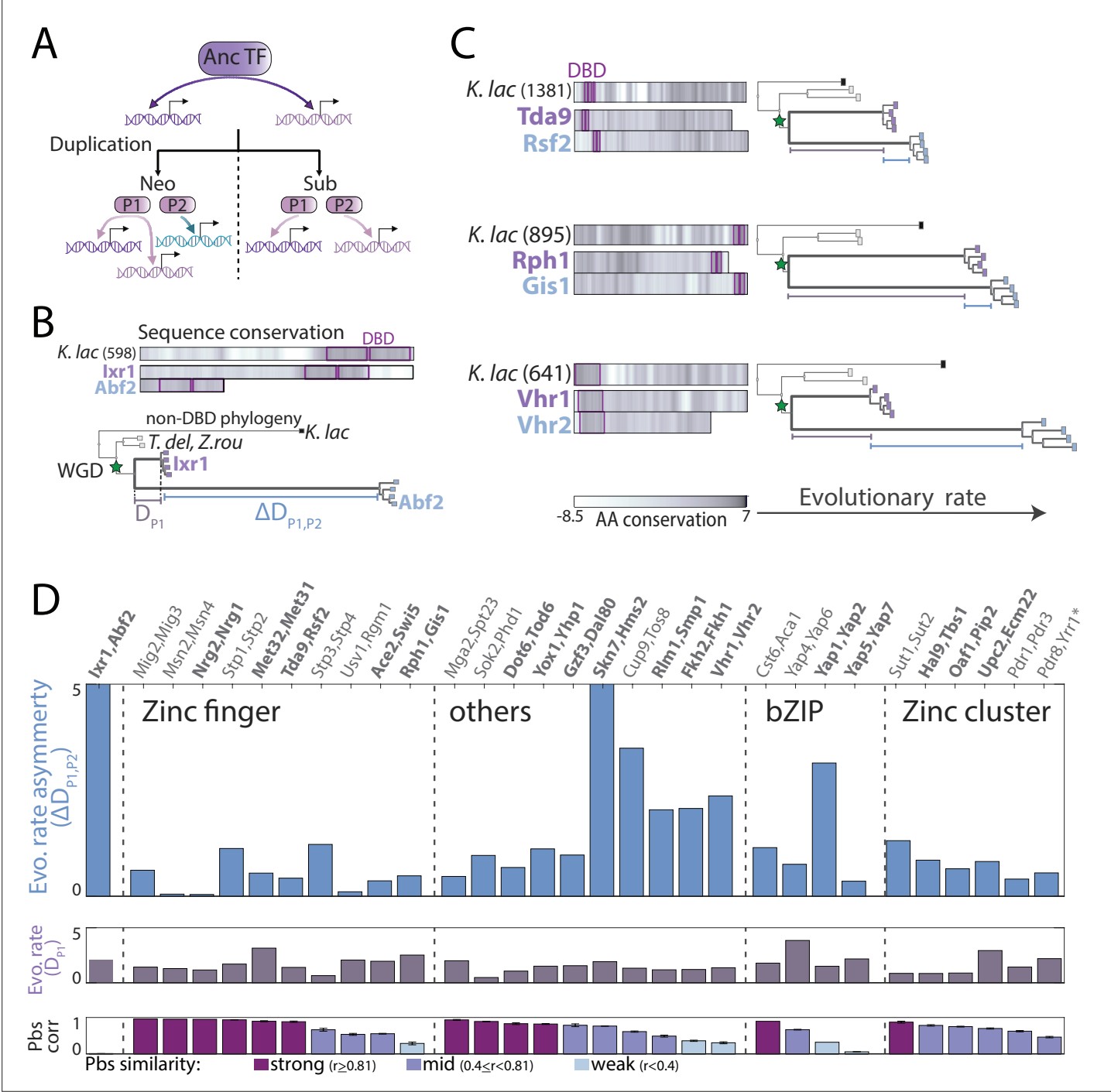

**Figure 6.** Asymmetric sequence evolution in whole-genome duplication (WGD) transcription factor (TF) paralog pairs. (**A**) *Models of functional divergence after WGD*: Paralogs could diverge by one-sided acquisition of new preferences (neo-functionalization) or by splitting ancestral preferences (sub-functionalization). (**B–C**) *Sequence evolution of indicated paralog pairs*. (**B**) Sequence variations among Ixr1/Abf2, a strongly diverged paralog pair. Top: Sequence conservation between the *Kluyveromyces lactis* ortholog and the non-WGD consensus sequence, or each *Saccharomyces cerevisiae* paralogs and the *K. lactis* ortholog along the respective protein length. Conservation score is the smoothened amino acid (AA) substitution score of the respective residue in a pairwise sequence alignment (see Materials and methods). Bottom: Phylogenetic comparison of non-DNA-binding domain sequences, indicating distance from the last common ancestor (LCA) to the conserved paralog (purple line, $D_{P1}$), and the distance difference between the paralogs, that is, evolutionary rate asymmetry (blue line, $\Delta D_{P1,P2}$, see Materials and methods for details). (**C**) As in (**B**) for the indicated paralog pairs with different levels of evolutionary rate asymmetry (see *Figure 6—figure supplement 1* for all pairs). (**D**) Evolutionary rate asymmetry ($\Delta D_{P1,P2}$),

*Figure 6 continued on next page*

*Figure 6 continued*

evolutionary rate of the conserved paralog (D$_{P1}$), and correlation in promoter-binding signals (pbs) for all paralog pairs. Paralogs chosen for further experimental analysis are highlighted in bold (*: lacking *K. lactis* ortholog).

The online version of this article includes the following figure supplement(s) for figure 6:

**Figure supplement 1.** Asymmetric sequence evolution in whole-genome duplication (WGD) transcription factor (TF) paralog pairs.

Extending the analysis to other diverging paralogs pointed at additional cases which might conform to this same design whereby limited binding competition resolved residual paralog interference (*Figure 9F* and *Figure 9—figure supplement 1*). Together, these results suggest a common path, whereby DBD-independent divergence is complemented by asymmetric competition, limiting residual paralog interference.

## Discussion

The binding of TFs at individual regulatory regions can vary through mutations that alter the DNA sequence or mutations that change TF-binding preferences. While promoter mutations are gene specific, changes in TF-binding preferences will affect multiple genes, and are therefore less likely to occur (*Teichmann and Babu, 2004*; *Hsia and McGinnis, 2003*; *Carroll, 2005*). TF duplication, which generates two redundant copies and relaxes selection, may provide an easy route for the evolution of binding preferences. The evolutionary paths through which binding preferences of TF paralogs diverge may therefore hold the key to understanding the principles that guide TF-binding site selection across the genome.

Studying a comprehensive set of WGD-retained TF paralogs, we found that the majority of pairs still share a large fraction of overlapping targets. In fact, even diverged paralogs still localized to common targets, although at different relative strengths. This gradual divergence was not explained by variations in the DBDs. In particular, we presented evidence that, with the exception of the zinc cluster family, variations within the DBD contributed little to binding divergence. DBD preferences may play a minor role in setting promoter-binding preferences but be primarily important for stabilizing binding within selected promoters. Further, cooperation and competition act to adjust binding profiles and limit paralog interference but, with few exceptions, are not the major factors guiding divergence. In this context, gradual, and promoter-specific divergence is harder to explain within models in which TF-binding specificity depends on a single DNA-binding motif or a single recruiting factor. In the case of a single recruiting factor, for example, we would expect an 'all-or-none' behavior that is common to a subset of genes. The gradual, gene-specific divergence we observe may be more naturally explained in models where binding depends on multiple specificity determinants within the TF, which recognize a corresponding multiplicity of features within the promoter. The existence of such multiplicity would allow to tune more readily TF binding at the level of individual promoters. We recently described one such paradigm in the context of the Msn2 and Yap1 TFs, whose promoter-binding depends on the cumulative contribution of a large number of specificity determinants distributed throughout their long (>500 amino acids) intrinsically disordered regions (*Jana et al., 2021*; *Brodsky et al., 2020*; *Brodsky et al., 2021*).

At the more global level, since duplication is the major means through which new TFs emerge, the evolutionary trajectories of retained paralogs shape the transcriptional network's design. Duplicates that diverge through sub-functionalization, for example, will confer a hierarchical design, where regulatory modules are gradually refined as the network expands. By contrast, neo-functionalization may support a distributed design, where new regulatory modules can emerge largely independent of previous connectivity. Focusing on the 60% diverging pairs, our results reveal that neo-functionalization is quite common, although it is often combined with a biased sub-functionalization, namely uneven splitting of ancestral targets.

Whereas we focused our analysis on diverging paralogs, it is notable that a significant fraction of paralog pairs (~40%) still binds at practically identical sites. Retention of these paralogs is therefore due to other properties. Duplicates' tendency to cross-bind their own promoters may suggest that interactions between duplicates have evolved to confer beneficial dynamic properties not achieved

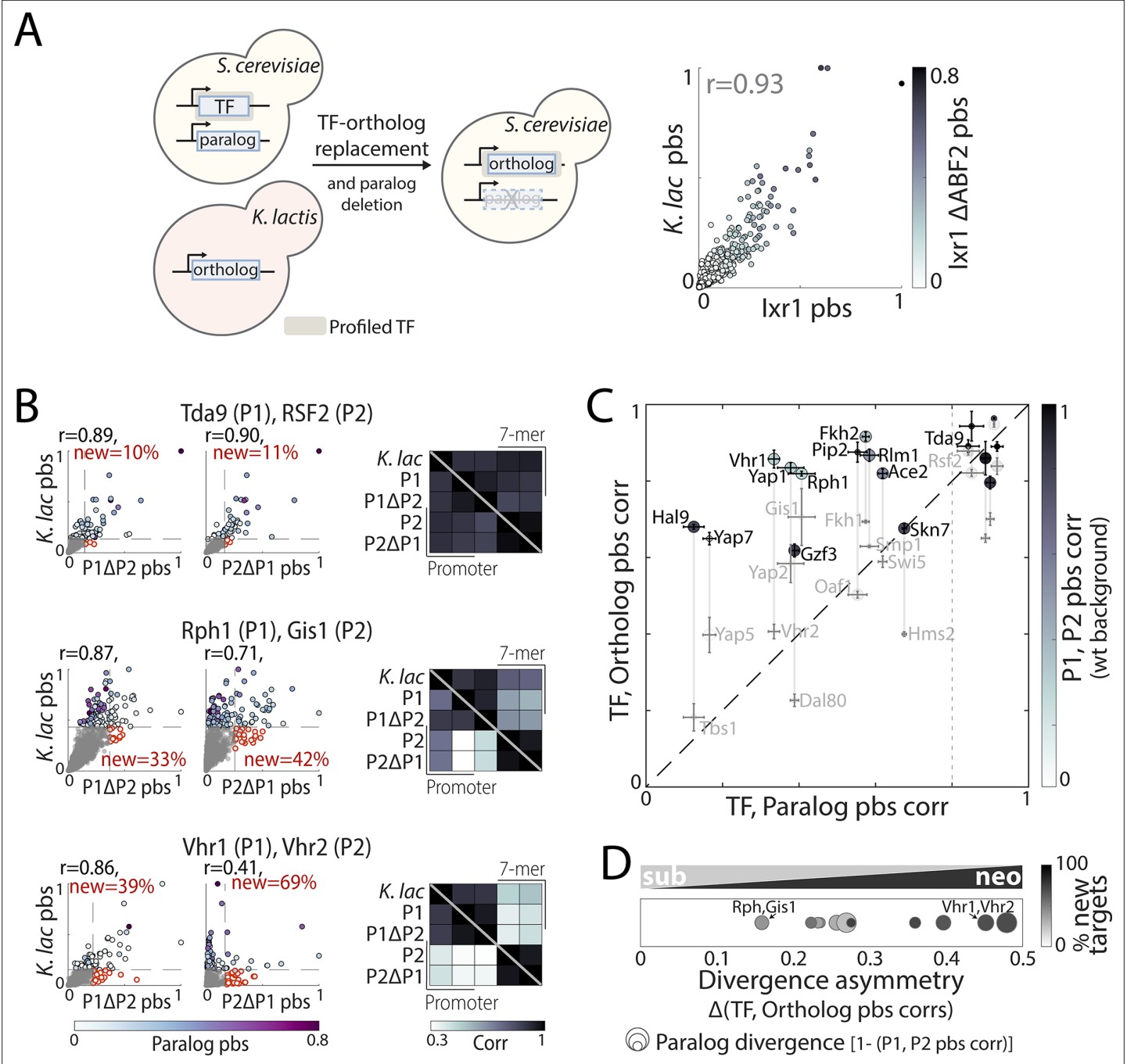

**Figure 7.** Evolution of binding preferences between *Kluyveromyces lactis* and *Saccharomyces cerevisiae* transcription factor (TF) orthologs. (**A–C**) *Mapping and comparing non-whole-genome duplication K. lactis ortholog binding profiles within S. cerevisiae*: Experimental scheme (left) and promoter-binding signal (pbs) for Ixr1/Abf2 *K. lactis* ortholog compared with *S. cerevisiae* Ixr1, in wild-type (x-axis) and ABF2-deletion background (color, right). (**B**) Pbs and correlations of binding preferences (bottom triangle: promoters, top triangle: 7-mers) between the *K. lactis* ortholog and *S. cerevisiae* paralogs in wild-type and paralog-deletion backgrounds, for the same example pairs shown in *Figure 6* (r: Pearson's correlation, red: percentage of new among strong targets). (**C**) For all paralog pairs with profiled orthologs, correlation between *S. cerevisiae* and *K. lactis* orthologs (y-axis) shown as a function of the correlation between *S. cerevisiae* paralogs (x-axis). Correlations were measured in paralog-deletion background, with paralogs' correlation in wild-type background shown in shade. Also shown are the sequence evolutionary rate (spot size; large spots reflect paralog with slower evolutionary rate, *Figure 6*) and difference in pbs correlation of the *S. cerevisiae* paralogs with their *K. lactis* ortholog (divergence asymmetry, defined as |corr(P1,ortholog)-corr(P2,ortholog)|, gray vertical lines). Note the strong similarity of binding preferences between each *K. lactis* TF with at least one of the *S. cerevisiae* paralogs, commonly the one experiencing slower sequence evolution. The dashed line indicates the divergence cut-off used in (**D**). (**D**) *Evolution through biased neo/sub-functionalization*: Diverged paralog pairs (with corr(P1, P2)<0.8 as indicated by dashed line in (**C**)) are positioned

*Figure 7 continued on next page*

*Figure 7 continued*

according to the divergence asymmetry of their correlation with the *K. lactis* ortholog (x-axis, (**C**)). Color indicates the percentage of new, among strong targets acquired by the less conserved paralog, and spot size indicates divergence of promoter-binding preferences between the paralogs (***Figure 7— figure supplement 1*** for all tested paralog pairs).

The online version of this article includes the following source data and figure supplement(s) for figure 7:

**Source data 1.** Details for statistical analysis (correlations).

**Figure supplement 1.** *Kluyveromyces lactis* orthologs represent possible binding preferences of the pre-duplication ancestor and suggest biased neo/ sub-functionalization as the dominant divergence principle.

by a single TF (***Teichmann and Babu, 2004***; ***Chapal et al., 2019***; ***Lehner, 2010***). Further studies are required to examine the potential advantages provided by such circuit-forming duplicates. Finally, we note that although TF binding is necessary for gene regulation, it is not sufficient. Hence, diverging binding preferences are not always translated into changes in transcriptional activity. Future studies may examine how the binding divergence described here affects transcription regulation.

# Materials and methods

## Key resources table

| Reagent type (species) or resource | Designation | Source or reference | Identifiers | Additional information |
|---|---|---|---|---|
| Strain, strain background (*Saccharomyces cerevisiae*) | BY4741 | PMID:9483801 | | |
| Strain, strain background (*Kluyveromyces lactis*) | CLIB209 | | | |
| Other | *S. cerevisiae* C-terminal SWAp-Tag (C-SWAT) | PMID:29988096 | | Yeast strain library (Background: BY4741) |
| Other | *S. cerevisiae* N-terminal SWAp-Tag (N-SWAT) | PMID:29988094 | | Yeast strain library (Background: BY4741) |
| Recombinant DNA reagent | bRA89 (plasmid) | PMID:28405019 | RRID:Addgene_100950 | |
| Recombinant DNA reagent | pGZ108 (plasmid) | PMID:26490019 | RRID:Addgene_70231 | |
| Software, algorithm | MATLAB | MathWorks | | |
| Software, algorithm | Bowtie 2.0 | PMID:30020410 | | |
| Software, algorithm | BEDTools | PMID:20110278 | | |
| Software, algorithm | cutAdapt | https://doi.org.10.14806/ej.17.1.200 | | |
| Software, algorithm | CHOPCHOP | PMID:24861617 | | |

## Strains and constructs

### Plasmids

All CRISPR transformations were performed using the bRA89 backbone plasmid (***Anand et al., 2017***), encoding Cas9, the target-specific guide-RNA and Hygromycin resistance. The target-specific spacer RNA template was designed using CHOPCHOP (***Labun et al., 2019***), ligated into the pre-cut bRA89 vector as previously described (***Anand et al., 2017***) and transformed into *Escherichia coli* for propagation. Plasmids were verified with PCR and purified with MiniPrep Kit (Real Genomics).

### Yeast

All genetic manipulations were performed in the *S. cerevisiae* BY4741 background (***Baker Brachmann et al., 1998***), with the MATa his3Δ1 leu2Δ0 met15Δ0 ura3Δ0 genotype. Transformations were performed using the LiAc/SS DNA/PEG method (***Gietz et al., 1995***). Following validation, the bRA89 plasmid from positive colonies was lost by growth in YPD (yeast extract peptone dextrose) and selection for colonies without bRA89-encoded Hygromycin resistance. Specific genotypes of all strains used in this study are listed in ***Supplementary file 3***.

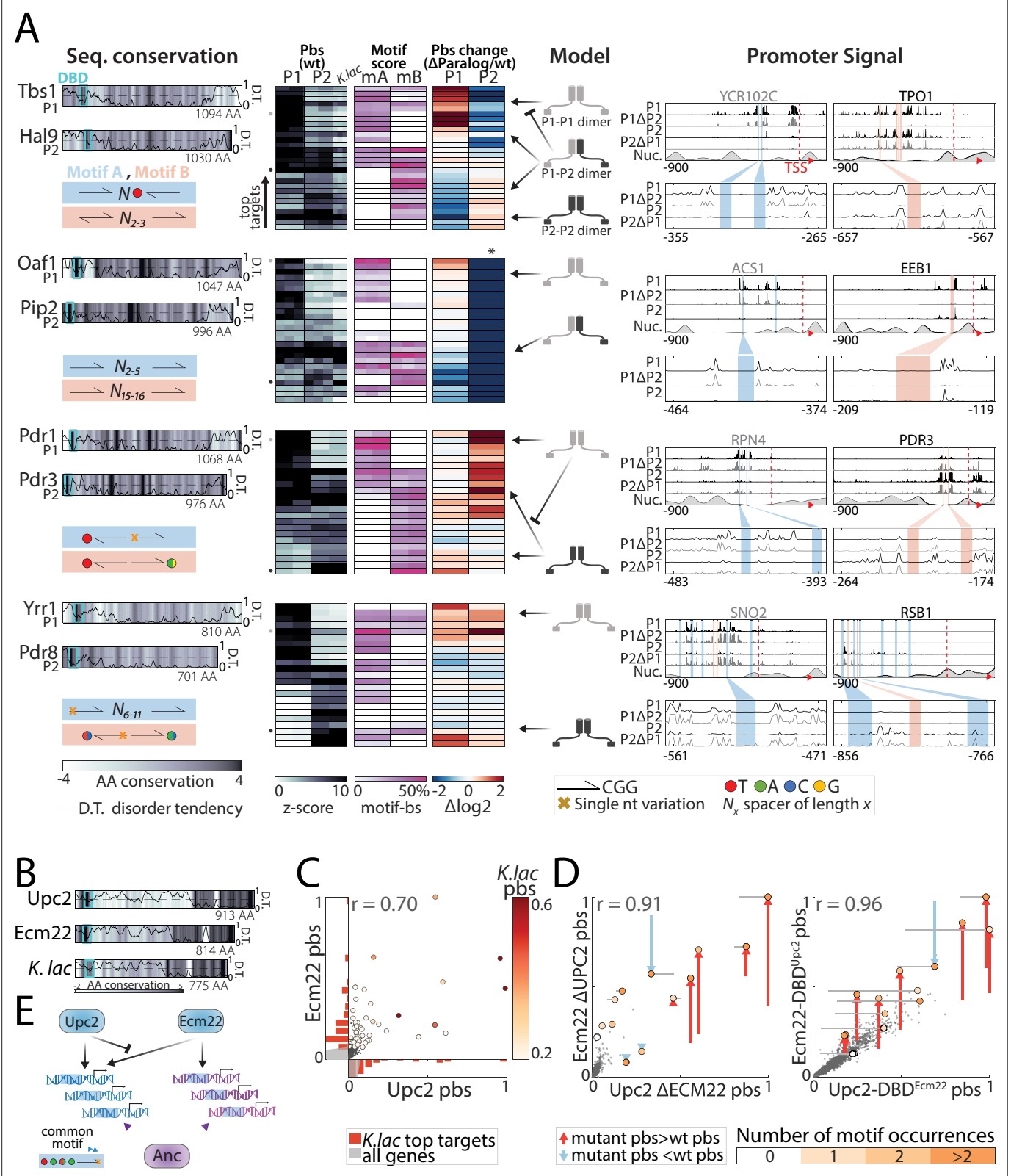

**Figure 8.** Divergence of zinc cluster transcription factor (TF) paralogs correlates with changes in motif preferences. (**A**) *Dimerization and changes in motif preferences may explain divergence of zinc cluster paralogs:* Zinc cluster paralogs vary in sequence and localized at different variants of their characteristic motif. Shown on the left with pairwise amino acid (AA) sequence conservation shown as color-code, DNA-binding domain (DBD) position indicated as cyan box, and disorder tendency (*Mészáros et al., 2018*) shown as black line; motif symbols indicated on the bottom (see *Figure 8—*

*Figure 8 continued on next page*

*Figure 8 continued*

*figure supplement 1* for motif sequences). For each pair, top-bound promoters were selected, and peak-proximal motifs defined. Shown, as indicated, are promoter-binding signal (pbs, z-score, columns correspond to individual repeats), percentage of total promoter signal 50 bases around the indicated motifs (columns correspond to individual repeats), and binding change upon paralog deletion (log2, mean; *: indicates loss of binding specificity after paralog deletion). Suggested models explaining divergence, and the signal on exemplary promoters (indicated by small gray and black dots next to the pbs panel) are also shown. (**B–E**) *Upc2/Ecm22 diverge through DNA-binding competition:* shown in (**B**) are the disorder tendency (*Mészáros et al., 2018*) and pairwise sequence conservation of Upc2-Ecm22 along the respective protein length and that of their *K. lactis* ortholog with Upc2 (*Figure 6—figure supplement 1*). Promoter-binding preferences in the indicated backgrounds are shown in (C–D). (**C**) Large dots indicate top 50 *K. lactis* targets, color-coded by binding signal. Distribution of these targets across the Upc2/Ecm22 binding preferences are shown as histograms (red, gray: all promoters). Note that Upc2 and Ecm22 bind comparably to strong *K. lactis* targets, while Ecm22 dominates on low-intermediate targets. (**D**) Large dots indicate Upc2 and Ecm22 top 20 targets (in wild-type background), colors indicate the number of occurrences of the known *in vitro* motif (TA(T/A)ACGA) and arrows show change in binding relative to the wild-type. (**E**) Suggested model: Ecm22 and Upc2 bind a common motif, but Upc2 outcompetes Ecm22 on Upc2's share of ancestral targets.

The online version of this article includes the following figure supplement(s) for figure 8:

**Figure supplement 1.** Homo- and heterodimerization's role in the binding preferences and divergence of dimer-forming transcription factor (TF) paralog pairs.

## Wild-type TFs tagged with micrococcal nuclease (MNase)

TF open reading frames (ORFs) were either C- or N-terminally tagged with MNase using the C-/N--terminal SWAp-Tag (C-SWAT, N-SWAT) libraries (*Meurer et al., 2018*; *Weill et al., 2018*) as parental strains. The SWAT acceptor module was replaced with MNase using CRISPR. Yeast cells were transformed with a repair template (PCR-amplified MNase coding DNA sequence from the pGZ108 plasmid; *Zentner et al., 2015*) and bRA89 plasmid with guide-RNA targeting the SWAT acceptor module. Colonies were confirmed using PCR and gel electrophoresis followed by DNA sequencing. Few strains were generated and profiled in previous studies from the lab (*Brodsky et al., 2020*; *Lupo et al., 2021*).

## DBD-swapping strains

DBD-Swapping strains were generated from the wild-type, MNase-tagged TF background strains, using CRISPR. The cells were transformed with a genomic PCR amplification product of the corresponding paralog's DBD sequence as repair template and a locus-specific bRA89 plasmid. Colonies were confirmed using PCR and gel electrophoresis followed by DNA sequencing. Used DBD annotations are shown in *Figure 4—figure supplement 1* and primers used to prepare repair amplicons are listed in *Supplementary file 4*.

## Paralog-deletion strains

Deletion strains were generated from the wild-type MNase-tagged TF background strains using homologous recombination of a PCR-amplified Kanamycin or Nourseothricin resistance cassette from the pBS7 (*Hailey et al., 2002*) or pFA6natNT2 (*Janke et al., 2004*) plasmids, respectively. Colonies were confirmed using PCR and gel electrophoresis.

## *K. lactis* ortholog gene replacement

*K. lactis* ortholog replacement strains were generated from the deletion strains, using CRISPR. The cells were transformed with a *K. lactis* (CLIB 209) genomic PCR amplification product of the corresponding ortholog gene sequence as a repair template and locus-specific bRA89 plasmid. The *K. lactis* gene was inserted to replace the MNase-tagged TF ORF, keeping the endogenous promoter and the MNase tag. Colonies were confirmed using PCR and gel electrophoresis followed by DNA sequencing. Primers used to prepare repair amplicons are listed in *Supplementary file 4*.

## ChEC-seq experiments

The experiments were performed as described previously (*Zentner et al., 2015*), with some modifications. Yeast strains were freshly thawed before experiments from a frozen stock, plated on YPD plates, and grown. Single colonies were picked and grown overnight at 30°C in liquid SD (synthetic complete with dextrose) medium to stationary phase. Then, the cultures were diluted ~2×10³-fold into 5 mL fresh SD media and grown overnight to reach an $OD_{600}$ of 4 the following morning. Cultures

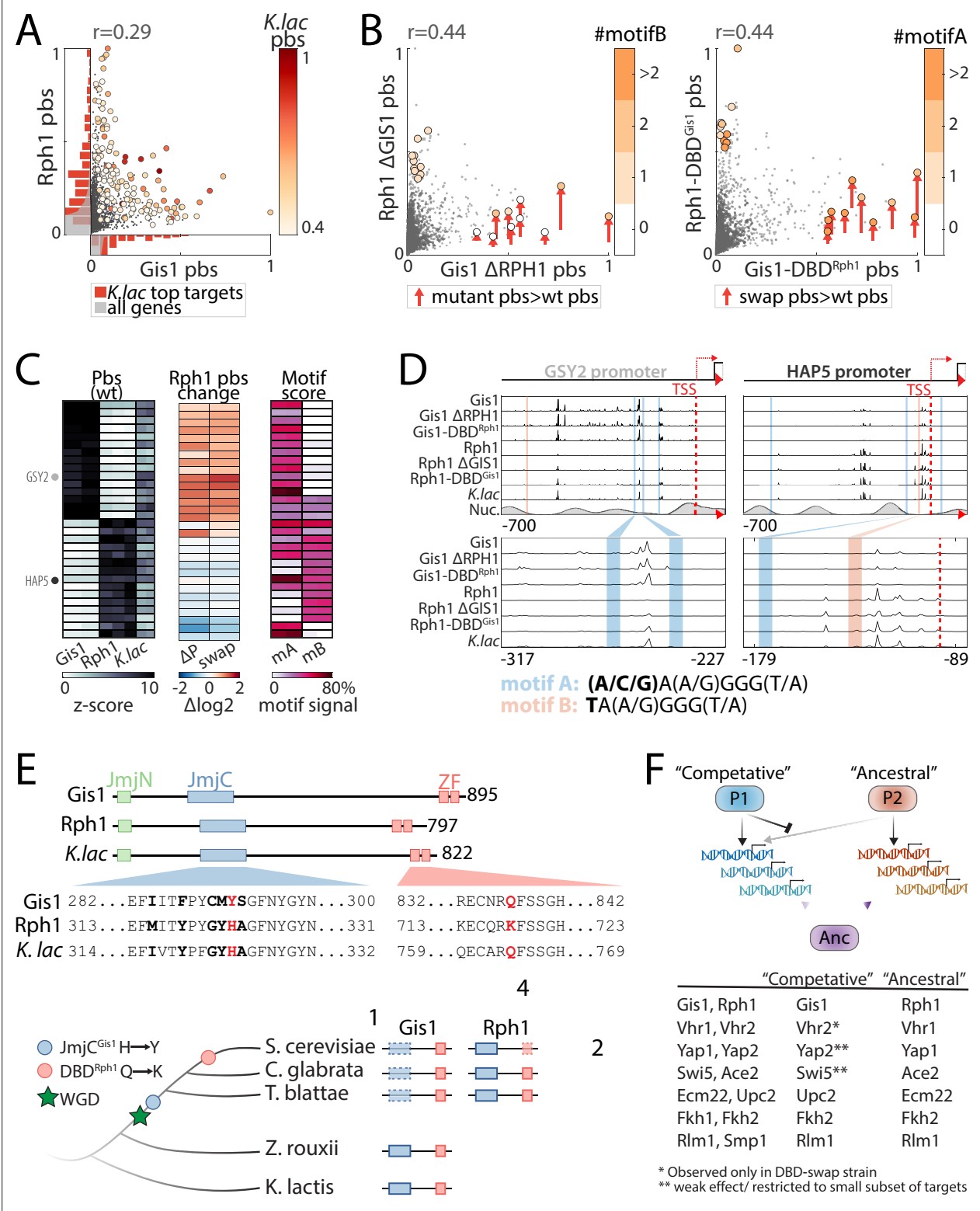

**Figure 9.** Resolution of paralog interference through competitive binding. (**A–D**) *Gis1 limits Rph1 binding through DNA-binding domain (DBD)-dependent competition:* shown are promoter-binding preferences of Gis1 and Rph1 in wild-type backgrounds (**A**, as in *Figure 8C*) and following mutual paralog deletion and DBD-swapping (**B**, colored by the number of occurrences of the two known motif variants specified in **D**). The analysis of all top-bound promoters is summarized in (**C**) (columns correspond to individual repeats) and binding signals on exemplary promoters are shown in (**D**) (as described in *Figure 8A*). Note the increased binding of Rph1 to Gis1 target promoters (e.g. GSY2) upon GIS1 deletion or DBD-swapping

*Figure 9 continued*

(in wild-type background), and reduced Gis1 binding to its target promoter after DBD-swapping (e.g. HAP5). (**E**) *Gis1's loss of demethylase activity preceded variation in Rph1's DBD*: The conserved JmjC domain providing Rph1 a histone demethylase activity is mutated in Gis1 orthologs of all post-WGD species. The respective DBDs differ in only four positions, at one of which a conserved glutamine is replaced by lysine specifically in Rph1 and its closest orthologs (***Figure 9—figure supplement 1***). This suggests that the divergence was triggered by the loss of demethylase function and DBD-independent acquisition of new targets by Gis1, and a final mutation in Rph1-DBD to reduce residual Rph1-binding interference at the newly acquired Gis1 sites (blue box: JmjC domain, red box: DBD, dashed box: mutated domain). (**F**) *Resolution of paralog interference among diverging transcription factor (TF) paralogs*: A model for the resolution of paralog interference through competitive binding. The TF inhibiting its paralog's binding is denoted as 'competitive', while the TF whose binding preferences better resemble those of the *Kluyveromyces lactis* ortholog is denoted as 'ancestral'. In addition to Gis1/Rph1, other diverging paralogs whose *K. lactis* orthologs were profiled appear to conform to this general model (***Figure 9—figure supplement 1***). Note that in most cases (indicated), divergence in promoter binding is driven by variations outside the DBD, with competition only refining, but not driving this divergence of target preferences.

The online version of this article includes the following figure supplement(s) for figure 9:

**Figure supplement 1.** The role of competition and paralog interference in the divergence of transcription factor (TF) paralogs.

were pelleted at 1500 *g* for 2 min and resuspended in 0.5 mL buffer A (15 mM Tris pH 7.5, 80 mM KCl, 0.1 mM EGTA, 0.2 mM spermine, 0.5 mM spermidine, 1× cOmplete EDTA-free protease inhibitors (Roche, 1 tablet per 50 mL buffer), 1 mM PMSF) and then transferred to 1 mL 96-well plates (Thermo Scientific). Cells were washed twice in 1 mL Buffer A. Next, the cells were resuspended in 150 µL Buffer A containing 0.1% digitonin, transferred to an Eppendorf 96-well plate (Eppendorf 951020401) and incubated at 30°C for 5 min for permeabilization. Next, we added CaCl$_2$ to a final concentration of 2 mM to activate the MNase and incubated for exactly 30 s. The MNase treatment was stopped by adding an equal volume of stop buffer (400 mM NaCl, 20 mM EDTA, 4 mM EGTA, and 1% SDS) to the cell suspension. After this, the cells were treated with Proteinase K (0.5 mg/mL) at 55°C for 30 min. An equal volume of Phenol-Chloroform pH = 8 (Sigma-Aldrich) was added, vigorously vortexed and centrifuged at 17,000 *g* for 10 min to extract DNA. After phenol chloroform extraction of nucleic acids, the DNA was precipitated with 2.5 volumes of cold 96% EtOH, 45 mg Glycoblue, and 20 mM sodium acetate at –80°C for>1 hr. DNA was centrifuged (17,000 *g*, 4°C for 10 min), supernatant removed and the DNA pellet washed with 70% EtOH. DNA pellets were dried and resuspended in 30 µL RNase A solution (0.33 mg/mL RNase A in Tris-EDTA [TE] buffer [10 mM Tris and 1 mM EDTA]) and treated at 37°C for 20 min. In order to enrich for small DNA fragments and remove large DNA fragments that might result from spontaneous DNA breaks, DNA cleanup was performed using SPRI beads (Ampure XP, Beckman Coulter). First, a reverse SPRI cleanup was performed by adding 0.8× (24 µL) SPRI beads followed by 5 min incubation at RT. Supernatant was collected and the remaining small DNA fragments purified by adding additional 1× (30 µL) SPRI beads and 5.4× (162 µL) isopropanol, and incubating 5 min at RT. Beads were washed twice with 85% EtOH and small fragments were eluted in 30 µL of 0.1× TE buffer.

## Next-generation sequencing library preparation

Library preparation was performed as described in ***Skene and Henikoff, 2017***, with slight modifications. DNA fragments after RNase treatment and reverse SPRI cleanup served as an input to end-repair and A-tailing (ERA) reaction. For each sample 20 µL ERA reaction (1× T4 DNA ligase buffer [NEB], 0.5 mM dNTPs, 0.25 mM ATP, 2.75% PEG 4000, 6U T4 PNK [NEB], 0.5U T4 DNA Polymerase [Thermo Scientific] and 0.5U Taq DNA polymerase [Bioline]) was prepared and incubated for 20 min at 12°C, 15 min at 37°C and 45 min at 58°C in a thermocycler.

After ERA reaction, reverse SPRI cleanup was performed by adding 0.5× (10 µL) SPRI beads (Ampure XP, Beckman Coulter). Supernatant was collected and remaining small DNA fragments purified with additional 1.3× (26 µL) SPRI beads and 5.4× (108 µL) isopropanol. After washing with 85% EtOH, small fragments were eluted in 17 µL of 0.1× TE buffer; 16.4 µL elution were taken into 40 µL ligation reaction (1× Quick ligase buffer [NEB], 4000U Quick ligase [NEB], and 6.4 nM Y-shaped barcode adaptors with T-overhang ***Blecher-Gonen et al., 2013***) and incubated for 15 min at 20°C in thermocycler.

After incubation, ligation reaction was cleaned by performing a double SPRI cleanup: first, a regular 1.2× (48 µL) SPRI cleanup was performed and eluted in 30 µL 0.1× TE buffer. Then and instead of separating the beads, an additional SPRI cleanup was performed by adding 1.3× (39 µL) HXN buffer (2.5 M NaCl, 20% PEG 8000) and final elution in 24 µL 0.1× TE buffer; 23 µL elution were taken into

50 µL enrichment PCR reaction (1× Kappa HIFI [Roche], 0.32 µM barcoded Fwd primer and 0.32 µM barcoded Rev primer *Blecher-Gonen et al., 2013*) and incubated for 45 s in 98°C, 16 cycles of 15 s in 98°C and 15 s in 60°C, and a final elongation step of 1 min at 72°C in a thermocycler.

The final libraries were cleaned by a regular 1.1× (55 µL) SPRI cleanup and eluted in 15 µL 0.1× TE buffer. Library concentration and size distribution were quantified by Qubit (Thermo Scientific) and TapeStation (Agilent), respectively. For multiplexed next-generation sequencing (NGS), all barcoded libraries were pooled in equal amounts, the final pool diluted to 2 nM and sequenced on NextSeq 500 (Illumina) or NovaSeq 6000 (Illumina). Sequence parameters were Read1: 51 nucleotides (nt), Index1: 8 nt, Index2: 8 nt, Read2: 51 nt, for NovaSeq or Read1: 38 nt, Read2: 37 nt for NextSeq.

## NGS data processing

Raw reads from ChEC-seq libraries were demultiplexed using bcl2fastq (Illumina), and adaptor dimers and short reads were filtered out using cutadapt (*Martin, 2011*) with parameters: '--O 10 –pair-filter = any –max-n 0.8 –action = mask'. Filtered reads were subsequently aligned to the *S. cerevisiae* genome R64-1-1 using Bowtie 2 (*Langmead and Salzberg, 2012*) with the options '--end-to-end --trim-to 40 --very-sensitive'. The genome coverage of fully aligned read pairs was calculated with GenomeCoverage from BEDTools (*Quinlan and Hall, 2010*) using the parameters '-d –5 –fs 1' resulting in the position of the fragment ends, which correspond to the actual MNase cutting sites. This was confirmed by the small median fragment size in the libraries (<150 bps), which is unlikely to result from spontaneous DNA breaks. All further processing of samples with more than 200,000 concordantly aligned reads or with >0.9 correlation (Pearson's r) among biological repeats was performed using MATLAB. First, the total coverage was normalized so that the mean coverage on the nuclear genome was one. Good repeats were selected based on internal correlation, and also used to generate the mean profile for each strain (at least two repeats per strain).

## Quantification and statistical analyses

### Promoter definition

Transcription start sites (TSS) were defined by combining publicly available TSS datasets (*Park et al., 2014*; *Pelechano et al., 2013*; *Policastro et al., 2020*). Promoter region was defined from the start codon until at least 700 bp upstream of the TSS (start codon if no TSS was available), or the closest verified ORF.

### Promoter-binding representation

For comparison of the normalized binding signal on specific promoter examples as shown in *Figures 1D, 5D, 8A and 9D*, signals were scaled so that the upper limit represents 50%, 50%, 20%, and 40% of the maximal signal height across the genome in each sample, respectively. Region of promoters shown is as follows: in *Figures 1C and 5D* 700 nt upstream to the start codon and 150 nt downstream into the ORF, same for *Figure 9D* but with 20 nt into the ORF. In *Figure 8A*, 900 nt upstream to the start codon and 100 nt downstream into the ORF. Nucleosome occupancy was taken from *Chereji et al., 2018* and smoothened with a Gaussian filter with STD = 25 nt.

### Promoter-binding quantification

Promoter-binding signal (pbs) was calculated by summing the normalized coverage over the promoter region of each gene (n = 5424). For comparison between different TFs, the pbs of each promoter by a certain TF was normalized to the promoter with the highest pbs for this TF.

### TF choice for profiling

After constructing MNase strains for 78 out of 82 TF paralogs, we decided to proceed only with those pairs for which: (a) both paralogs could be successfully profiled under the conditions used and (b) both paralogs mostly bind to promoters of specific target genes. We therefore excluded Rsc3/Rsc30*, Aft1/Aft2*, Haa1/Cup2*, Itc1/YPL216W*, Vid22/Env11*, and Nfi1/Siz1* where at least one paralog could not be profiled reliably or does not show sequence-specific TF activity (indicated by an asterisk), as well as Nhp6A/Nhp6B, which displayed no clear target preference. Reb1/Nsi1 or Ixr1/Abf2 were excluded as one paralog did not localize to promoter regions but ribosomal DNA or to the mitochondria genome, respectively (see *Supplementary file 1*).

## Significant TF promoter binding for regulatory circuit analysis (Figure 1E)

Significant TF promoter binding was defined by z-score threshold at the 99% quantile but not more than 3.5.

## Correlation between pbs or 7-mer binding signal of two samples (Figures 2–9)

To determine the similarity of binding signals (promoter or 7-mer) between strain A and strain B, we calculated the Pearson's correlation between each individual repeat of strain A with each individual repeat of strain B, that is, $n_{repeatsA} * n_{repeatsB}$ correlations in total. Then we calculated and show the mean and standard deviation (error bars) of Pearson's r (*Figure 2*, *Figure 3B*, *Figure 4C*, *Figure 5B-C*, *Figure 6D* and *Figure 7C*). When plotting the pbs of one strain against the pbs of another strain (*Figures 1D, 3A, 5D, 7A–B, 8C-D and 9A–B*), the mean signals are used.

## Visualizing binding changes in scatter plots (Figures 8C and 9B, Figure 8— figure supplement 1 and Figure 9—figure supplement 1)

In order to define the binding changes, the promoter signals in the mutant strains (DBD-swap or paralog deletion) are adjusted so that the mean signal of the top10 wild-type promoters in the mutant strain is the same as their mean signal in the wild-type strain.

## Relative, gene-specific binding changes upon paralog deletion or DBD-swapping (Figures 5, 8 and 9)

ChEC-seq only reports on the relative, but not absolute, binding strength along the genome, and due to the normalization a decreased binding to some targets will increase the relative signal at non-targets. To measure gene-specific changes in binding signal, we assumed similar 'absolute' binding at most targets based on the strong binding correlation between the mutants and their corresponding wild-types (see *Figure 5*). We then adjust the normalized pbs and use these adjusted values to compare the binding changes of the other targets: first, a robust linear regression (MATLAB function: robustfit) between the wild-type (independent variable) and the mutant pbs across the 50 strongest bound promoters (or more, if z-score>3.5) was performed. The slope of this fit was then used to adjust the mutant pbs: $pbs_{adjusted} = pbs_{mutant}/slope$ and the adjusted value compared to the wild-type binding ($pbs_{wt}$): $log2((pbs_{adjusted} + 700)/(pbs_{WT} + 700))$. Significantly changing genes were defined as genes, whose relative binding change exceeded the mean of the 50 strongest bound promoters by at least one STD. For the actual plots in *Figure 5—figure supplement 1*, we show the binding changes of the top 40 targets of each paralog pair that are also shown in *Figure 1* and are listed in *Supplementary file 2*.

## Pip2 and Hms2 DNA binding depends on the presence of their paralogs (Figures 5 and 8, Figure 8—figure supplement 1)

DNA-binding profiles of Pip2 and Hms2 in the absence of their paralogs, Oaf1 and Skn7 respectively, could not be obtained. At least four biological repeats of each strain showed extremely low correlations of promoter binding (mean Pearson's r of 0.03 or 0.011 in four or five biological repeats for Pip2 and Hms2, respectively, data not shown). In addition, none of the repeats showed similarity with the wild-type strain (Pearson's r < 0.25, 0.42 for Pip2 and Hms2, respectively, data not shown). Pip2 acting primarily as a heterodimer with Oaf1 is supported in the literature (*Rottensteiner et al., 1997*).

## 'New' strong targets determination for neo/sub-functionalization classification of *S. cerevisiae* paralogs (Figure 7, Figure 7—figure supplement 1)

To focus on the evolutionary history of strong *S. cerevisiae* targets, targets of the *S. cerevisiae* paralogs and *K. lactis* ortholog were defined based on a pbs z-score threshold of 4.5 and 3.5, respectively. For each *S. cerevisiae* paralog, the percentage of targets not among the *K. lactis* targets was defined as 'new'.

## 7-mer binding signal quantification

Each genomic position was indexed according to the 7-mer sequence surrounding it (–3 to +3) with assigning the same index to forward and reverse complement sequences (8192 indexes in total). Considering the properties of ChEC-seq (MNase cutting in the vicinity of the binding site, but protection of the actual binding site by the TF), the ChEC-seq signal, representing the actual cutting sites, was processed with a filter that subtracts the 7 nt moving average from the 21 nt moving average for each position and thereby punishes cutting sites inside the respective 7-mer. Negative values were set to zero and the mean binding score for each 7-mer index was calculated from the processed signal in promoter regions (excluding ORFs).

## Mean *in vivo* signal around *in vitro* motif occurrences

Position weight matrixes (PWMs) of *in vitro* motifs (obtained by protein-binding microarrays) were downloaded from CISBP (*Weirauch et al., 2014*). In the case of more than one available PWM, all *in vitro* PWMs of this paralog pair were compared using correlation distance. The PWM couple with the highest correlation was chosen as the PWMs for this paralog pair. For better comparability between the TFs, only the most probable bases at the five most informative positions were used (in-between bases were replaced by N) to define a simplified motif. The whole genome was scanned for exact matches to these simplified motifs using regular expressions, from these found matches only the matches inside promoter regions were kept for further analysis. The binding signal 300 nt around each occurrence was extracted. As shown in *Figure 3A* for the 3302 and 5471 *in vitro* motif occurrences in promoters for Gis1/Rph1 and Dot6/Tod6, respectively. The mean signal for these 300 nt windows centered on the selected occurrences was calculated (*Figure 3—figure supplement 1*). These simplified motifs were also used to select the *in vitro* motif-containing *in vivo* 7-mers (*Figure 3A*).

## DBD sequence comparison between paralogs

For each paralog pair, DBD sequences based on Pfam DBD positions (*Mistry et al., 2021*), determined using hmm-scan, were extracted and aligned using hmm-align (*Madeira et al., 2019*). If the domain was part of the similarity regression (SR) analysis (*Lambert et al., 2019*), the conserved amino acid residues were defined as every residue with >50% of the maximal conservation score. Specificity-conferring residues were defined as every residue with an SR score >150% of the mean SR score in this domain. For the paralog pairs without SR-analyzed DBD domain; Spt23/Mga2 (TIG), Rlm1/Smp1 (SRF), and Vhr1/Vhr2 (Vhr1), conservation score was obtained from Pfam domain HMMs using Skylign (*Wheeler et al., 2014*) and used to determine conserved amino acids residues like above. Specificity-defining residues were not determined for these paralog pairs (see *Figure 3*, *Figure 3—figure supplement 1*). To distinguish between functionally similar and different substitutions, amino acids were classified into positively charged (Lys, Arg, His), negatively charged (Asp, Glu), hydrophilic (Thr, Ser, Asn, Gln, Cys, Tyr), and hydrophobic (Ala, Trp, Val, Ile, Leu, Pro, Phe, Met).

## Phylogeny analysis of non-DBD sequences

For each paralog pair, all ortholog sequences were downloaded from YGOB (*Byrne and Wolfe, 2005*) and their Pfam-based DBD positions determined using hmm-scan (*Madeira et al., 2019*). Non-DBD sequences (after removing the DBDs) were aligned using m-coffee (*Moretti et al., 2007*) with the options '-method t_coffee_msa clustalo_msa mafft_msa muscle_msa kalign_msa clustalw2_msa pcma_msa'.

To construct the maximum likelihood tree from a constrained input tree, the non-DBD sequence alignments were then used as an input for IQTree (*Nguyen et al., 2015*) with ultra-fast bootstrapping (*Hoang et al., 2018*), options: '-m JTT + I + G4+ Fbb 1000g inputtree -wsr -asr -redo'. The constrained input tree was based upon known species phylogeny (see *Figure 6—figure supplement 1*) and distinguished between the *K. lactis*/*Eremothcium* clade, the *Zygosaccharomyces rouxii*/*Torulaspora delbrueckii* clade, the *Lachancae* clade, and all post-WGD paralogs. To adjust for protein-specific differences in evolution rates, all distances on the calculated tree were normalized so that the mean distance between *T. delbruecki* and *Z. rouxii* to their last common ancestor was 1. These normalized values are presented in *Figure 6B and C* and *Figure 6—figure supplement 1*. For visualization, the trees were subsequently simplified by removing all leaves (and branches) belonging to species other than the *Sacchormyces strictu* clade, *K. lactis*, *Z. rouxii*, and *T. delbrueckii*.

## Protein sequence conservation along protein length

To compare the sequence conservation between two protein sequences, for example, two orthologs, global sequence alignment with the BLOSUM62 scoring matrix between these two proteins was performed, and the derived substitution scores were mapped back onto the corresponding residues in the original protein sequence. The conservation score was then calculated as the 20-residues moving average of this substitution score. In *Figure 6*, *Figure 8* and *Figure 6—figure supplement 1* the sequences of both *S . cerevisiae* paralogs are compared to their *K. lactis* ortholog (or *Z. rouxii* if *K. lactis* was not available). The *K. lactis* ortholog was either compared to Upc2 (*Figure 8*) or the non-WGD ortholog sequence consensus (*Figure 6* and *Figure 6—figure supplement 1*) derived from multiple sequence alignment of the full-length proteins with m-coffee as described above.

## Acknowledgements

We like to thank Sagie Brodsky, Tamar Jana, and Offir Lupo for their strains and all Barkai lab members for fruitful scientific discussions and careful reading of the manuscript.

## Additional information

### Competing interests

Naama Barkai: Senior editor, *eLife*. The other authors declare that no competing interests exist.

### Funding

| Funder | Grant reference number | Author |
| --- | --- | --- |
| Israel Science Foundation | | Tamar Gera<br>Felix Jonas<br>Roye More<br>Naama Barkai |
| Minerva Foundation | | Tamar Gera<br>Felix Jonas<br>Roye More<br>Naama Barkai |
| H2020 European Research Council | | Tamar Gera<br>Felix Jonas<br>Roye More<br>Naama Barkai |

The funders had no role in study design, data collection and interpretation, or the decision to submit the work for publication.

### Author contributions

Tamar Gera, Felix Jonas, Conceptualization, Data curation, Formal analysis, Investigation, Methodology, Software, Validation, Visualization, Writing - original draft, Writing - review and editing; Roye More, Investigation; Naama Barkai, Conceptualization, Funding acquisition, Project administration, Resources, Supervision, Writing - original draft, Writing - review and editing

### Author ORCIDs

Tamar Gera ⬤ http://orcid.org/0000-0002-8216-2411
Felix Jonas ⬤ http://orcid.org/0000-0001-7214-8942
Naama Barkai ⬤ http://orcid.org/0000-0002-2444-6061

### Decision letter and Author response

Decision letter https://doi.org/10.7554/eLife.73225.sa1
Author response https://doi.org/10.7554/eLife.73225.sa2

## Additional files

### Supplementary files

• Supplementary file 1. Whole-genome duplication (WGD)-generated transcription factor (TF) paralog pair selection. List of all DNA-binding domain (DBD)-containing WGD-generated paralogs in *Saccharomyces cerevisiae* with DBD family and the performed experiments (gray font indicates filtered-out paralog pairs).

• Supplementary file 2. Top targets for each transcription factor (TF) paralog pair (compare *Figure 2*). For each pair (sorted by family and inter-pair Pearson's r) the top 40 targets based on promoter-binding signal (as used in *Figure 2*) are listed with their standard name, systematic name, and the promoter-binding signal (z-score) by each paralog of the pair.

• Supplementary file 3. Yeast strains used in this study. List of strains used in this study with their background genotype and source. (In the genotype-column 'TF' stands for the open reading frame (ORF) of the MNase-tagged TF and 'tf' for that of its paralog.)

• Supplementary file 4. Primers used to prepare DNA-binding domain (DBD) swap variants and *Kluyveromyces lactis* strains. For each created DBD-swap/ortholog replacement strain, the forward and reverse primers used in the genomic PCR (*Saccharomyces cerevisiae* genome for DBD-swaps and *K. lactis* genome for ortholog replacement, respectively) are listed.

• Transparent reporting form

### Data availability

Sequencing data have been deposited in GEO under accession codes GSE179430. MATLAB scripts for analysis and visualization are available on GitHub (https://github.com/barkailab/Gera2021; copy archived at swh:1:rev:863f900e6fd11d761005eca01ce1725c953dfa25). Figure 1—source data 1, Figure 4—source data 1, Figure 5—source data 1, and Figure 7—source data 1 contain the numerical data used to generate the summary figures.

The following datasets were generated:

| Author(s) | Year | Dataset title | Dataset URL | Database and Identifier |
|---|---|---|---|---|
| Gera T, Jonas F | 2021 | WGD paralog evolution | https://www.ncbi.nlm.nih.gov/geo/query/acc.cgi?acc=GSE179430 | NCBI Gene Expression Omnibus, GSE179430 |
| Gera F, Jonas F, More R, Barkai N | 2022 | Evolution of binding preferences among whole-genome duplicated transcription factors | https://doi.org/10.5061/dryad.xgxd254j6 | Dryad Digital Repository, 10.5061/dryad.xgxd254j6 |

The following previously published dataset was used:

| Author(s) | Year | Dataset title | Dataset URL | Database and Identifier |
|---|---|---|---|---|
| Brodsky J | 2020 | Intrinsically disordered regions direct transcription factor in-vivo binding specificity | https://www.ncbi.nlm.nih.gov/bioproject/?term=PRJNA573518 | NCBI BioProject, PRJNA573518 |

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
