## [Editor Report]

The authors use creative and innovative approaches to explore the evolution of transcription factors following duplication. This paper will be broad interest among evolutionary and molecular biologists as it addresses the long-standing question of how newly evolved transcription factor proteins acquire new binding specificities or split ancestral ones. Among the major findings, the authors find the changes in binding specificity occur mainly through changes outside of the DNA binding domains, showing that novelties in transcription factor binding can occur through multiple routes following gene duplications.

---

## [Decision Letter]

**Decision letter after peer review:**

Thank you for submitting your article "Evolution of binding preferences among whole-genome duplicated transcription factors" for consideration by *eLife*. Your article has been reviewed by 3 peer reviewers, and the evaluation has been overseen by a Reviewing Editor and Kevin Struhl as the Senior Editor. The following individual involved in review of your submission has agreed to reveal their identity: Elena Kuzmin (Reviewer #1).

Essential revisions:

The three reviewers provided detailed comments that would need to be addressed, they may require significant editing and addition of details regarding the analyses and the writing.

1) Transcription factor binding does not translate directly into changes in gene expression. This could be discussed and addressed in the manuscript.

2) Some of the figures are heavily loaded with results and some labels appear to be missing. Some statistical tests are not fully reported. Please see how this can be corrected.

3) One of the reviewers mentioned that the manuscript is difficult to read in some sections, which is in line with reviewer 2's comments regarding missing information for some plots and statistics. I would therefore recommend reviewing the manuscript accordingly.

4) The justifications of some cutoff values in the analyses are not provided. Please provide a clear justification each time.

*Reviewer #1 (Recommendations for the authors):*

Gera et al., seek to understand the factors that lead to duplicated gene divergence focusing on transcription factors in *S. cerevisiae*. They tested DNA binding preferences of 30 pairs of transcription factors originating from whole genome duplication event. ChEC-seq was used to profile DNA binding preferences. Authors experimentally swapped DNA binding motifs of 15 duplicates to test if changes in the DNA binding motif led to the divergence of paralogs' motif preferences. Promoter binding preferences showed little variation since 3 of 15 (80%) paralogs didn't show any variation leading to the conclusion that variation in motif preferences was due to variation encoded outside of the DNA binding domain. DNA binding of each paralog was tested in the background of WT or deletion background of second paralogs to find the prevalence of cooperative or competitive protein interactions between paralogs. Cooperative interactions were less prevalent than competitive interactions. Authors analyzed variation outside DNA binding domain to detect paralogs that diverged by sub- or neo-functionalization. They found that C2H2 zinc-finger family showed evidence of symmetric evolution consistent with subfunctionalization compared to other families that evolved asymmetrically, indicative of neofunctionalization. They experimentally tested subfunctionalization and neofunctionalization scenarios by replacing a *S. cerevisiae* with a K. lactis homolog as a proxy for the ancestral gene and found that most paralogs show evidence of both fates terming this process biased sub/neofunctionalization whereby paralogs split their binding preferences unevenly while the acquisition of new binding targets mostly occurs for the most divergent paralog. By focusing on zinc-cluster family, they identified pairs with diverged binding preferences due to DNA motif preferences and dimerization with their paralog. They proposed that zinc-cluster family of transcription factors divergence is influenced by the acquiring distinct preferences but restricted by the competitive binding of the other paralog what they refer to as 'limiting paralog interference'. This is an elegant study, and the manuscript is a pleasure to read!

*Reviewer #2 (Recommendations for the authors):*

Page 2 "Dimer-forming paralogs evolved mostly one-sided dependency, while other paralogs interacted through low-magnitude DNA-binding competition that minimized paralog interference." – This sentence is very hard to understand without reading the manuscript before. Once I had read the manuscript, I did understand, but it is cryptic if one comes across this sentence in the abstract before reading the manuscript. I would recommend either expanding and explaining it better or reducing it to a more general, understandable minimum.

Figure 1C: What is shown on the axis? "Gzf3" (for example) is not a measurement, nor does it show a unit. The same is the case for similar plots throughout the paper (2A, 3D, 4D, 5C, 6A, several suppl figures)

Figure 2A top example: How do you interpret that at promoters it's anticorrelated but at 7-mers it's correlated?

The triangles in Figure 2D are not explained well. Why are there four horizontal boxes but only two x axis labels (Gis1,Rph1 in the left example)? I understood it once I looked at Figure S2D, but it should be clear already here.

Similar in Figure 2E, what are those axis labels supposed to mean? Do I interpret those correctly as correlations of the swapped vs unswapped DBD (y axis) and correlations of the swapped vs unswapped rest of the protein (x axis)? Given that dots represent comparisons between two TFs, what comparison is denoted by the labels? "Gis1" is not comparing Gis1 to itself, but Gis1 to a domain-swapped TF, but which?

Figure 3 "Tendency for paralog interactions varies between TF families" Is this supported by the data? All three plots look strikingly similar, with most paralogs showing no substantial difference, a few some difference and the occasional one-offs with very large effects. The latter are n=1 cases, so can't make a story out of it.

Figure 4C The experimental scheme is not clear at all and should be explained better. Was one paralog replaced by a K. lactis ortholog and the other one deleted in every measurement?

Figure 4E The figure axis labels are very hard to understand. Which axes include K. lactis data? Both?

Page 8 "zinc-cluster paralogs […] were more likely to interact (cf. Figure 3)" I do not see how that is supported by the data presented in Figure 3

Figure 5C "lines show change in binding relative to the wild-type" How is this to be read? Which end is WT, which is mutant?

Figure 5D This model should be explained more explicitly.

Page 10 "Evolution of binding preferences, however, is possible through TF duplication," It may also be possible under other circumstances. I would tone down this phrase.

Page 11 "may be more consistent with the paradigm of distributed specificity we recently proposed" It would be good to briefly define/explain this here instead of just referencing papers. Why do the authors think this is the case? I can imagine a number of alternative explanations for this finding.

It is not described how exactly in vitro PWMs were mapped. I would recommend using specific PWM mapping software for in silico mapping of in vitro PWMs, like FIMO or many others.

Statistical reporting (transparent reporting form): A lot of statistical tests were performed. For instance, every correlation is a statistical test and has an associated P value that is never stated anywhere in the manuscript.

*Reviewer #3 (Recommendations for the authors):*

The manuscript is very interesting asking an excellent question and using elegant approaches to investigate the impact of duplications on transcription factor binding site evolution.

Although the results are sounds the manuscript (text and figures) needs significant editing to make it much clearer and easier to read.

---

## [Author Response]

Reviewer #2 (Recommendations for the authors):Page 2 "Dimer-forming paralogs evolved mostly one-sided dependency, while other paralogs interacted through low-magnitude DNA-binding competition that minimized paralog interference." – This sentence is very hard to understand without reading the manuscript before. Once I had read the manuscript, I did understand, but it is cryptic if one comes across this sentence in the abstract before reading the manuscript. I would recommend either expanding and explaining it better or reducing it to a more general, understandable minimum.

We changed this sentence, which now reads:

"Interactions between paralogs were rare, and, when present, occurred through weak competition for DNA-binding or dependency between dimer-forming paralogs."

Figure 1C: What is shown on the axis? "Gzf3" (for example) is not a measurement, nor does it show a unit. The same is the case for similar plots throughout the paper (2A, 3D, 4D, 5C, 6A, several suppl figures)

The figure shows the cumulative binding signal of indicated TF (e.g. Gzf3) along the promoter and relative to the promoter with the highest binding signal. We thank the reviewer for noting that our labeling is not clear. We now add the measure, pbs (promoter binding signal), and explain this better in the caption and method.

Figure 2A top example: How do you interpret that at promoters it's anticorrelated but at 7-mers it's correlated?

This is exactly our point about binding specificity: the two TFs bind the same sequence motif (localize to the same 7-mers, hence high correlation); however, this motif is highly abundant in the genome and only a fraction is bound. Each of these factors then ‘chose’ to bind a different subset of these motifs, found in different promoters (hence low correlation in promoter signal). We included an additional plot showing the signal around all motif occurrences (new Figure 3A).

The triangles in Figure 2D are not explained well. Why are there four horizontal boxes but only two x axis labels (Gis1,Rph1 in the left example)? I understood it once I looked at Figure S2D, but it should be clear already here.

We changes the labeling in Figure 2D (new Figure 4B), adopting the style of Figure S2D (adding additional y-labels and making the title clearer).

Similar in Figure 2E, what are those axis labels supposed to mean? Do I interpret those correctly as correlations of the swapped vs unswapped DBD (y axis) and correlations of the swapped vs unswapped rest of the protein (x axis)? Given that dots represent comparisons between two TFs, what comparison is denoted by the labels? "Gis1" is not comparing Gis1 to itself, but Gis1 to a domain-swapped TF, but which?

We changed the labeling and annotation in this Figure and added schematics to explain the measures. Moreover, we also improved the annotation of distinct dots to show the actual chimera, e.g. Gis1-DBD^Rph1^ that is compared with the DBD-paralog and the nonDBD-paralog.

Figure 3 "Tendency for paralog interactions varies between TF families" Is this supported by the data? All three plots look strikingly similar, with most paralogs showing no substantial difference, a few some difference and the occasional one-offs with very large effects. The latter are n=1 cases, so can't make a story out of it.

1. We agree that the numbers are too small to make general conclusions. We therefore removed the sentence.

2. To still emphasize the differences we observe between the families, we highlighted zinc cluster TFs in the middle-figure, which included both zinc cluster and bZip TFs. In this presentation, it is rather clear that the zinc cluster show substantially more interactions than most other paralogs, and in particular compared to the zinc finger TFs, where interactions are almost absent*.*

Figure 4C The experimental scheme is not clear at all and should be explained better. Was one paralog replaced by a K. lactis ortholog and the other one deleted in every measurement?

We thank the reviewer for noting that. We now expanded the schematic and its explanation (new Figure 6) to clarify the experimental approach.

Figure 4E The figure axis labels are very hard to understand. Which axes include K. lactis data? Both?

Only the y-axis includes *K. lactis* (ortholog) data, the x-axis only shows the correlation between the paralogs. We changed the presentation and labeling of Figure 4E (old, new Figure 7C). Note that it is now more similar to Figure 3B (new Figure 5B): x-label: TF, paralog pbs correlation; y-label: TF, orthologue pbs correlation. (pbs: promoter binding signal).

Page 8 "zinc-cluster paralogs […] were more likely to interact (cf. Figure 3)" I do not see how that is supported by the data presented in Figure 3

See answer to previous comment on Figure 3D.

Figure 5C "lines show change in binding relative to the wild-type" How is this to be read? Which end is WT, which is mutant?

We added an arrow indicating the direction from the wild-type to the mutant.

Figure 5D This model should be explained more explicitly.

We added a short explanation of this model to the figure caption:

“We conclude that Ecm22 and Upc2 bind a common motif, but Upc2 outcompetes Ecm22 on Upc2’s share of ancestral targets.”

Page 10 "Evolution of binding preferences, however, is possible through TF duplication," It may also be possible under other circumstances. I would tone down this phrase.

We agree and changed this to:

“TF duplication, which generates two redundant copies and relaxes selection, may provide an easy route for the evolution of binding preferences”

Page 11 "may be more consistent with the paradigm of distributed specificity we recently proposed" It would be good to briefly define/explain this here instead of just referencing papers. Why do the authors think this is the case? I can imagine a number of alternative explanations for this finding.

We expanded on our proposal:

“In this context, gradual, and promoter-specific divergence is harder to explain within models in which TF binding specificity depends on a single DNA binding motif or a single recruiting factor. In the case of a single recruiting factor, for example, we would expect an ‘all-or-none’ behavior that is common to a subset of genes. The gradual, gene-specific divergence we observe may be more naturally explained in models where binding depends on multiple specificity determinants within the TF, which recognize a corresponding multiplicity of features within the promoter. The existence of such multiplicity would allow to tune more readily TF binding at the level of individual promoters. We recently described one such paradigm in the context of the Msn2 and Yap1 TFs, whose promoter binding depends on the cumulative contribution of a large number of specificity determinants distributed throughout their long (>500 amino acids) intrinsically disordered regions (IDRs)”.

It is not described how exactly in vitro PWMs were mapped. I would recommend using specific PWM mapping software for in silico mapping of in vitro PWMs, like FIMO or many others.

We added a detailed description of how PWM were mapped in Materials and methods:

“We scanned the genome for exact matches to these simplified motifs using regular expressions, from these matches only those inside promoter regions were kept for further analysis”

We did not use FIMO since we were interested in shorter motif for this analysis (5-mers and not 8-mers): a large fraction of the TFs in our set is known to have a 5-mer motif, e.g. C2H2 zinc fingers. We verified that using FIMO with a p-value threshold of 0.001 and 0.0001 yields very similar results to our simplified motif scan.

Statistical reporting (transparent reporting form): A lot of statistical tests were performed. For instance, every correlation is a statistical test and has an associated P value that is never stated anywhere in the manuscript.

P-values and repeat numbers for each correlation are now stated in a source data table accompanying the figures.